# Maternal LSD1/KDM1A is an essential regulator of chromatin and transcription landscapes during zygotic genome activation

Katia Ancelin[1,2]*, Laurène Syx[1,3,4], Maud Borensztein[1,2], Noémie Ranisavljevic[1,2], Ivaylo Vassilev[1,3,4], Luis Briseño-Roa[5], Tao Liu[6], Eric Metzger[7], Nicolas Servant[1,3,4], Emmanuel Barillot[1,3,4], Chong-Jian Chen[6], Roland Schüle[7], Edith Heard[1,2]*

[1]Institut Curie, Paris, France; [2]Genetics and Developmental Biology Unit, INSERM, Paris, France; [3]Bioinformatics and Computational Systems Biology of Cancer, INSERM, Paris, France; [4]Mines ParisTech, Fontainebleau, France; [5]High Fidelity Biology, Paris, France; [6]Annoroad Gene Technology Co., Ltd, Beijing, China; [7]Urologische Klinik und Zentrale Klinische Forschung, Freiburg, Germany

**Abstract** Upon fertilization, the highly specialised sperm and oocyte genomes are remodelled to confer totipotency. The mechanisms of the dramatic reprogramming events that occur have remained unknown, and presumed roles of histone modifying enzymes are just starting to be elucidated. Here, we explore the function of the oocyte-inherited pool of a histone H3K4 and K9 demethylase, LSD1/KDM1A during early mouse development. KDM1A deficiency results in developmental arrest by the two-cell stage, accompanied by dramatic and stepwise alterations in H3K9 and H3K4 methylation patterns. At the transcriptional level, the switch of the maternal-to-zygotic transition fails to be induced properly and LINE-1 retrotransposons are not properly silenced. We propose that KDM1A plays critical roles in establishing the correct epigenetic landscape of the zygote upon fertilization, in preserving genome integrity and in initiating new patterns of genome expression that drive early mouse development.

*For correspondence: Katia. Ancelin@curie.fr (KA); Edith. Heard@curie.fr (EH)

**Competing interests:** The authors declare that no competing interests exist.

## Introduction

Gametes are highly differentiated cell types and fertilization of the oocyte by sperm requires major epigenetic remodelling to reconcile the two parental genomes and the formation of a totipotent zygote. In particular, the paternal genome arrives densely packed with protamines rather than histones, and the maternal epigenome is highly specialised. Maternal factors must unravel these specialised chromatin states to enable zygotic gene activation and development to proceed. Histone tail post-translational modifications (PTMs), and more specifically lysine methylation, appear to be dynamically regulated during this first step of development (*Burton and Torres-Padilla, 2010*). Histone lysine methylation appears to have different biological read-outs, depending on the modified residue as well as the state of methylation (mono-, di- or tri-). For example, methylation of histone H3 lysine 4 (referred to as H3K4 methylation hereafter) is mainly associated with transcriptionally active chromatin while methylation of histone H3 lysine 9 (referred to as H3K9 methylation hereafter) is usually linked to repressive chromatin. The incorrect setting of some of these histone marks in cloned animals have been correlated with their poor development potential, pointing to their importance during early stages of development (*Matoba et al., 2014*; *Santos et al., 2003*). However, the

**eLife digest** During fertilization, an egg cell and a sperm cell combine to make a cell called a zygote that then divides many times to form an embryo. Many of the characteristics of the embryo are determined by the genes it inherits from its parents. However, not all of these genes should be "expressed" to produce their products all of the time. One way of controlling gene expression is to add a chemical group called a methyl tag to the DNA near the gene, or to one of the histone proteins that DNA wraps around.

Soon after fertilization, a process called reprogramming occurs that begins with the rearrangement of most of the methyl tags a zygote inherited from the egg and sperm cells. This dynamic process is thought to help to activate a new pattern of gene expression. Reprogramming is assisted by "maternal factors" that are inherited from the egg cell.

KDM1A is a histone demethylase enzyme that can remove specific methyl tags from certain histone proteins, but how this affects the zygote is not well understood. Now, Ancelin et al. (and independently Wasson et al.) have investigated the role that KDM1A plays in mouse development.

Ancelin et al. genetically engineered mouse eggs to lack KDM1A and used them to create zygotes, which die before or shortly after they have divided for the first time. The zygotes display severe reprogramming faults (because methyl tags accumulate at particular histones) and improper gene expression patterns, preventing a correct maternal-to-zygotic transition. Further experiments then showed that KDM1A also regulates the expression level of specific mobile elements, which indicates its importance in maintaining the integrity of the genome.

These findings provide important insights on the crucial role of KDM1A in establishing the proper expression patterns in zygotes that are required for early mouse development. These findings might help us to understand how KDM1A enzymes, and histone demethylases more generally, perform similar roles in human development and diseases such as cancer.

actors underlying these dynamic changes in histone modifications after fertilization and their impact on the appropriate regulation of zygotic genome function remain open questions.

Lysine methylation is tightly regulated by distinct families of conserved enzymes, histone lysine methyltransferases (KMTs), which add methyl groups and histone lysine demethylases (KDMs) which remove them (*Black et al., 2012*). Importantly, KMTs and KDMs show different specificities for their target lysine substrates, as well as for the number of methyl group they can add or remove (from unmethylated -me0-, to dimethylated –me2, and trimethylated -me3; and vice versa).

Several KMTs and KDMs have been disrupted genetically in model organisms, including mouse, and their loss often leads to lethality or to severe defects in embryogenesis, or else in tissue-specific phenotypes in adults. This has been linked to their important roles in cell fate maintenance and differentiation, as well as in genome stability (*Black et al., 2012*; *Greer and Shi, 2012*). However, investigating their potential roles during the first steps of development, after fertilization is frequently hampered by their maternal mRNA and/or protein pool, which can persist during early embryogenesis and mask the potential impact that the absence of such factors might have (*Li et al., 2010*). In mice, conditional knock-outs in the female germline that suppress the maternal store of mRNA and protein at the time of fertilization, can be used to examine protein function during the earliest steps of development (*de Vries et al., 2000*; *Lewandoski et al., 1997*). In this way, the roles of KMTs and KDMs during early embryogenesis are just starting to be explored. For example, it has been shown that depletion of maternal EZH2 affects the levels of H3K27 methylation in zygotes, although this did not lead to any growth defects during embryonic development (*Erhardt et al., 2003*; *Puschendorf et al., 2008*). Another study investigated maternal loss of *Mll2* (Mixed lineage leukemia 2), encoding one of the main KMTs targeting H3K4 and revealed its essential role during oocyte maturation and for the embryos to develop beyond the two-cell stage, through gene expression regulation, (*Andreu-Vieyra et al., 2010*). Importantly, in the presence of maternal EZH2 or MLL2 protein (when wt/- breeders are used), both *Ezh2* and *Mll2* null embryos die much later *in utero* (*O'Carroll et al., 2001*; *Glaser et al., 2006*). The roles of these regulators of lysine methylation

can thus be highly stage-specific, with very different effects at the zygote, early cleavage or later developmental stages.

The LSD1/KDM1A protein (encoded by the gene previously known as *Lsd1* but subsequently renamed *Kdm1a,* which will be the used in this manuscript hereafter) was the first histone KDM to be characterized to catalyse H3K4me1 and 2 demethylation and transcriptional repression (*Shi et al., 2004*). KDM1A was later shown to demethylate H3K9me2 and to activate transcription (*Laurent et al., 2015*; *Metzger et al., 2005*). Genetic deletion of murine *Kdm1a* during embryogenesis obtained by mating of heterozygous animals showed early lethality prior to gastrulation (*Foster et al., 2010*; *Macfarlan et al., 2011*, *Wang et al., 2007*; *2009*). In light of the above considerations, we set out to study the impact of eliminating or inhibiting the maternal pool of KDM1A during preimplantation development. We report for the first time the crucial role of *Kdm1a* following fertilization. The absence of KDM1A protein in zygotes derived from *Kdm1a* null oocytes led to a developmental arrest at the two-cell stage, with a severe and stepwise accumulation of H3K9me3 from the zygote stage, and of H3K4me1/2/3 at the two-cell stage. These chromatin alterations coincide with increased perturbations in the gene expression repertoire, based on single embryo transcriptomes, leading to an incomplete switch from the maternal to zygotic developmental programs. Furthermore, absence of KDM1A resulted in deficient suppression of LINE-1 retrotransposon expression, and increased genome damage, possibly as a result of increased LINE-1 activity. Altogether, our results point to an essential role for maternally-inherited KDM1A in maintaining appropriate temporal and spatial patterns of histone methylation while preserving genome expression and integrity to ensure embryonic development beyond the two-cell stage.

## Results

### Depletion of maternal KDM1A protein results in developmental arrest at two-cell stage

To investigate whether *Kdm1a* might have a role during early mouse development we first assessed whether the protein was present in pre-implantation embryos using immunofluorescence (IF) and western blotting (*Figure 1A and B*). A uniform nuclear localization of KDM1A within both parental pronuclei was observed by IF in the zygote, and at the two-cell stage. The protein was also readily detected by western blot analysis of total extracts of two-cell-stage embryos when compared to nuclear extracts of ESCs. Altogether, these data reveal the presence of a maternal pool of KDM1A.

To assess the function of KDM1A in early mouse embryo development, we deleted the *Kdm1a* gene in the female germline during oocyte growth. To this end *Kdm1a*[tm1Schüle] *Zp3*[cre] females, carrying a new conditional allele for *Kdm1a* deletion engineered in the Schüle group (*Zhu et al., 2014*), and a *Zp3* promoter driven *cre* transgene exclusively expressed in oocytes (*Lewandoski et al., 1997*) were produced (see also materials and methods). These animals are referred as *Kdm1a*[f/f]:: *Zp3*[cre] in this study). *Kdm1a*[f/f]::*Zp3*[cre] females were then mated with wild-type males (*Figure 1C*). We isolated one- and two-cell stage embryos derived from such crosses to obtain maternally depleted *Kdm1a* mutant embryos (hereafter named △m/wt) in parallel to control embryos (hereafter named f/wt) and we confirmed that the KDM1A maternal pool is absent by performing IF (*Figure 1A*, bottom panels). In parallel, RT-qPCR analysis revealed the absence of *Kdm1a* mRNA in mutant oocytes (*Figure 1—figure supplement 1A*).

Numerous *Kdm1a*[f/f]::*Zp3*[cre] females were housed with wild-type males for several months, however no progeny was ever obtained, in contrast to *Kdm1a*[f/f] or [f/wt] females that produced the expected range of pup number (4 to 7; data not shown). This indicated that *Kdm1a*[f/f]::*Zp3*[cre] females are sterile. To determine the possible causes of sterility, control *Kdm1a*[f/f] and mutant *Kdm1a*[f/f]:: *Zp3*[cre] females were mated with wild-type males and embryos were recovered on embryonic day 2 (E2) (*Figure 1D and E*). The total number of oocytes or embryos scored per female was on average 17 for the mutant background (206 oocytes or embryos obtained for 12 females studied) and 25 for the control (226 oocytes or embryos obtained for 9 females studied) (see *Figure 1D*). We found that the proportion of △m/wt two-cell stage embryos recovered (19%, n = 39/206) was far lower than that obtained with f/wt embryos (75% n = 170/226) (*Figure 1D*). Using *Kdm1a*[f/f]::*Zp3*[cre] females, we also noted a high percentage of fertilized and unfertilized oocytes blocked at meiosis II (MII) (n = 95; 46%) compared to those recovered from control females (n = 34; 15%). Inspection of control

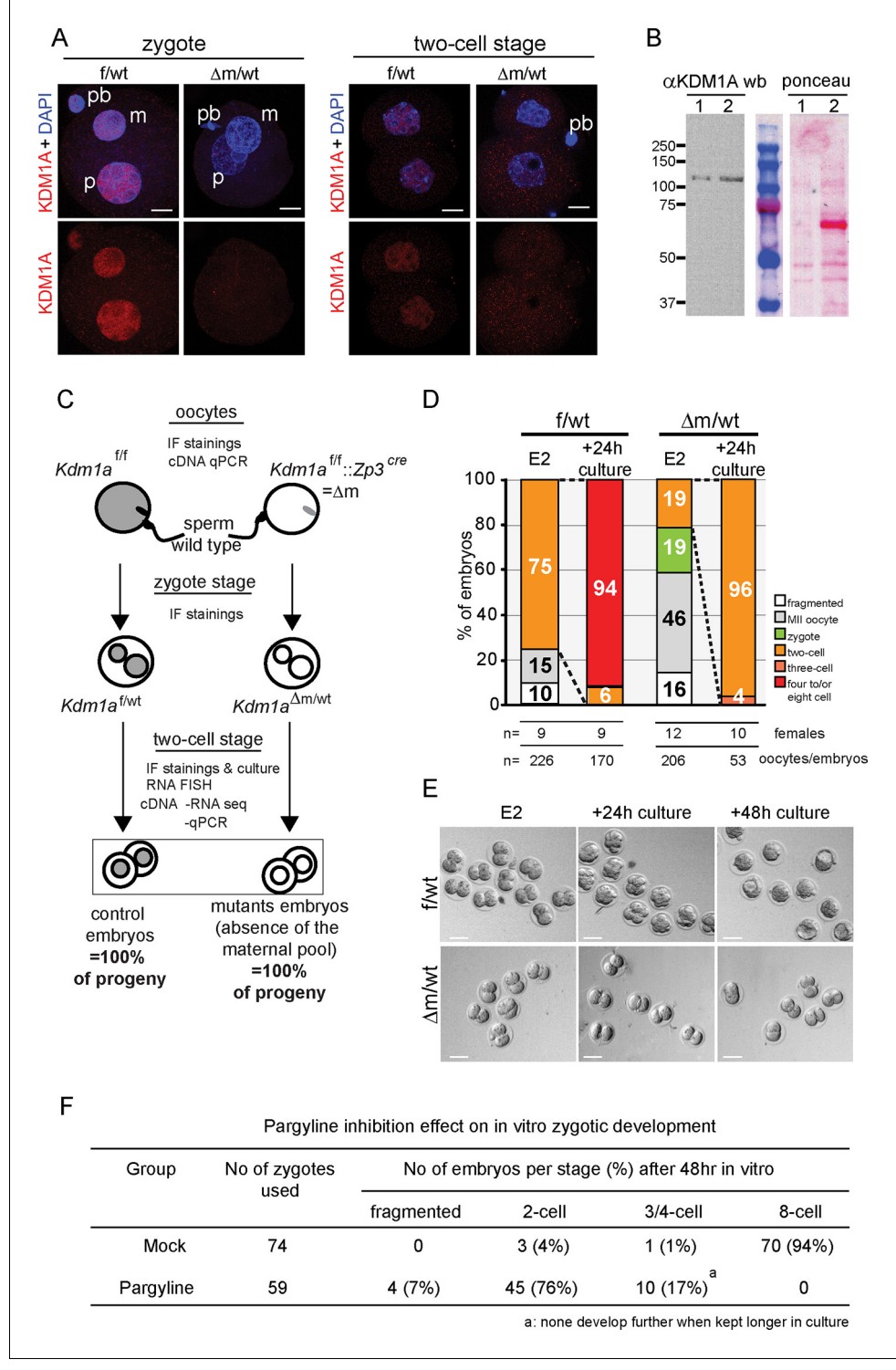

**Figure 1.** *Kdm1a* maternally deleted embryos arrest at two-cell stage. (**A**) Immunofluorescence using anti-KDM1A antibody (red) at the zygote and two-cell stage shows nuclear accumulation of KDM1A in control embryos (top). Cre-mediated deletion of *Kdm1a* in maternal germline (bottom) leads to depletion of the protein after fertilization. Paternal pronucleus (p), maternal pronucleus (m) and polar body (pb) are indicated. DNA is counterstained by DAPI (blue). (**B**) western blot analysis (left panel) for ESC (lane1) and two-cell stage embryo extracts (lane 2) using anti-KDM1A antibody. Ponceau staining (right panel) is shown as loading control. Molecular weights (kDa) are indicated on the left. (**C**) Mating scheme and experimental outcomes for the different developmental stages used in this study: f/wt control embryos are obtained from superovulated *Kdm1a*^f/f females mated with wild-type males, *Figure 1 continued on next page*

*Figure 1 continued*

while △m/wt mutant embryos are obtained from superovulated *Kdm1a*[f/f]::Zp3[cre] females crossed with wild-type males. (D) Distribution of developmental stages found in f/wt and △m/wt embryos collected at embryonic day 2 (E2) (expected two-cell stage) and after 24 hr of *in vitro* culture. Numbers of females used and numbers of oocytes/embryos analysedare shown under the graph. See also *Figure 1—figure supplement 1* for oocyte analysisand *Figure 1—figure supplement 2* for developmental stage distribution using natural matings without superovulation for females. (E) Bright field images representative for two consecutive days of in vitro culture for f/wt and △m/wt embryos collected at E2. (F) Phenotypes and distribution of developmental stages obtained after 48 hr treatment *in vitro* culture with a catalytic inhibitor (pargyline) of KDM1A in wild-type zygotes recovered at 17 hr post hCG injection. Scale bars represent, 10 μm and 50 μm, in A and D, respectively.

The following figure supplements are available for figure 1:

**Figure supplement 1.** *Kdm1a* loss of function in female germline.

**Figure supplement 2.** Embryo recovery at day 2 post fertilization using natural matings.

---

(n = 75) and mutant (n = 55) MII oocytes revealed a high proportion of misaligned chromosomes on the metaphase spindle (*Figure 1—figure supplement 1B and C*) in mutants (41%) compared to controls (17%), suggesting that a lack of maternal KDM1A can lead to chromosome segregation defects. Furthermore, upon fertilization, transmission of inherited chromosomal abnormalities was clearly evident, with the frequent presence of micronuclei in KDM1A maternally depleted two-cell embryos (n = 40; 63%) (*Figure 1—figure supplement 1D*). Lastly, 19% (n = 39) of mutant embryos were still at the zygote stage compared to 0% in controls (*Figure 1D*). These results indicate that many MII oocytes lacking germline KDM1A are not competent at ovulation and that when fertilized their first cell cycle is delayed. Similar results were obtained when using females not subjected to superovulation for mating (*Fig1—figure supplement 2*).

We next assessed the progress of surviving △m/wt two-cell embryos by culturing them in vitro. After 24 hr in culture, 96% of the △m/wt embryos were found to be arrested at the two-cell stage, unlike f/wt embryos where only 6% showed an arrest (*Figure 1D and E*). The mutant embryos blocked at the two-cell stage did not progress further in development upon prolonged in vitro culture, and eventually fragmented, while the control embryos progressed towards the blastocyst stage (*Figure 1E*).

Taken together, these results suggest that the sterility of *Kdm1a* germline mutant females is in part caused by a severely compromised spindle organization in some oocytes in the second round of meiosis,as well as for the second round of cleavage after fertilization. This immediate loss of viability of the first generation embryos contrasts with the progressive effect seen across generations when spr-5, the *Kdm1a* homologue in *C.elegans* is mutated in germline precursors for both gametes (*Katz et al., 2009*). Also, targeted disruption of Lsd2/*Kdm1b,* the closest homologue of *Kdm1a,* in the mouse female germline, was reported to have no effect on oogenesis and early mouse development, but only later at mid-gestation, due to misregulation at some imprinted genes (*Ciccone et al., 2009*). Our results show that KDM1B in the female germline is not sufficient to rescue the phenotype of KDM1A maternal depletion after fertilization.

## Inhibition of the enzymatic activity of KDM1A from early zygote stage mimics the maternal deletion phenotype

The developmental arrest observed at the two-cell stage of △m/wt embryos could be due to defects carried over by the mutant oocytes, particularly given the chromosome defects observed in a significant proportion of arrested oocytes, and/or to a requirement for KDM1A function after fertilization. To assess a requirement for KDM1A enzymatic activity in early embryos, we tested the impact of KDM1A catalytic inhibition specifically after fertilization. To this end, we treated wild-type zygotes with pargyline, a well-characterized potent chemical inhibitor of KDM1A enzymatic activity (*Fierz and Muir, 2012*; *Metzger et al., 2005*) and followed their development *in vitro* over 48 hr. As shown in *Figure 1F*, 76% embryos cultured with pargyline were found to be significantly blocked at the two-cell stage and 17% never progressed beyond the 3/4-cell stage. On the contrary, the

majority (94%) of mock treated embryos developed to the eight-cell stage within 48 hr, as expected. These data parallel the phenotype of genetic ablation of the KDM1A maternal pool, where 96% of △m/wt embryos are developmentally arrested at the two-cell stage and 4% at the 3/4-cell stage. Taken together, the genetic depletion and pargyline inhibition data strongly support a requirement for KDM1A enzymatic activity during the zygote and two-cell stage, for embryos to proceed beyond the two-cell stage.

## Abnormal increase of H3K9me3 levels in KDM1A maternally deficient zygotes

The above observations suggested that the histone demethylase KDM1A plays an important role in early development. At the zygote stage, H3K4 and H3K9 methylation levels appear to be tightly regulated and show highly parental specific patterns (*Arney et al., 2002*; *Lepikhov and Walter, 2004*; *Santos et al., 2005*; *Puschendorf et al., 2008*; *Santenard et al., 2010*; *Burton and Torres-Padilla, 2010*). Given that KDM1A has been implicated in the regulation of H3K4 and H3K9 mono and di methylation in previous studies (*Shi et al., 2004*; *Metzger et al., 2005*; *Di Stefano et al., 2008*; *Katz et al., 2009*), we investigated whether the methylation levels of these two histone H3 lysines were affected by KDM1A depletion in one-cell stage embryos.

To this end, we collected f/wt and △m/wt embryos at embryonic day 1 and analysed them for both H3K4 and H3K9 methylation using specific antibodies against mono (me1), di (me2) and tri (me3) methylation (*Figure 2*). We used antibodies that show similar patterns in control zygotes to those previously published by others (*Arney et al., 2002*; *Lepikhov and Walter, 2004*; *Puschendorf et al., 2008*; *Santenard et al., 2010*; *Santos et al., 2005*) (see Material and Methods). We prioritised single-embryo analysis given the limited amount of material that can be recovered at these early developmental time points, particularly in the context of the depletion of KDM1A (*Figure 1D*). We first analysed H3K4 methylation patterns (*Figure 2A and B*). It was previously reported that the paternal pronucleus only gradually shows enrichment in H3K4me2 and me3 during the one-cell stage, while the female pronucleus is enriched with these marks from its oocyte origin (*Burton and Torres-Padilla, 2010*; *Lepikhov and Walter, 2004*). We compared maternal and paternal pronuclear patterns in control and mutant embryos and categorised them according to previously described nomenclature (*Adenot et al., 1997*). In mid-stage zygotes, the absence of maternal KDM1A does not seem to affect overall H3K4me1, me2 or me3 levels in either the maternal or paternal pronuclei (*Figure 2A and B*).

We also assessed whether H3K9 methylation levels were affected in zygotes lacking a maternal pool of KDM1A (*Figure 2C and D*). H3K9me1 was reported to be equally enriched in both parental pronuclei, while H3K9me2 and me3 are exclusively present in the maternal pronucleus (*Arney et al., 2002*; *Santos et al., 2005*; *Lepikhov and Walter, 2004*; *Puschendorf et al., 2008*; *Santenard et al., 2010*; *Burton and Torres-Padilla, 2010*). We found that △m/wt embryos do not seem to differ from f/wt embryos in H3K9me1 levels (*Figure 2C and D*). In the case of H3K9me2, a complete absence of H3K9me2 staining in the paternal pronucleus was recorded for both control and mutant zygotes. However, we did note a small change in the proportion of embryos displaying H3K9me2 staining in the maternal pronucleus. This suggests that absence of KDM1A may slightly impact on oocyte-inherited H3K9me2 profiles.

Although KDM1A was shown to specifically induce demethylation of H3K9me1/2 at target genes (*Laurent et al., 2015*; *Metzger et al., 2005*), we nevertheless assayed H3K9me3 patterns by IF, in case it could also accumulate in absence of KDM1A, due to the presence of specific H3K9 KMT (*Cho et al., 2012*; *Puschendorf et al., 2008*). H3K9me3 enrichment is a feature of constitutive heterochromatin, and has been shown to be zygotically enriched at the periphery of nucleolar like bodies (NLBs) within the maternal but not the paternal pronucleus (*Burton and Torres-Padilla, 2010*; *Puschendorf et al., 2008*; *Santenard et al., 2010*) (*Figure 2C*). In △m/wt zygotes, strikingly elevated levels H3K9me3 were found in the whole maternal pronucleus when compared to controls (grey arrowhead, *Figure 2C and D*). Even more surprisingly, in △m/wt zygotes, H3K9me3 could be detected at the periphery of paternal NLBs (yellow arrowhead), when compared to controls. Taken together, these observations show that the absence of maternal KDM1A protein results in specifically elevated levels of H3K9me3 in both parental genomes at the zygote stage, and suggest that KDM1A might be engaged with other chromatin modifiers to regulate H3K9me3 immediately after fertilization.

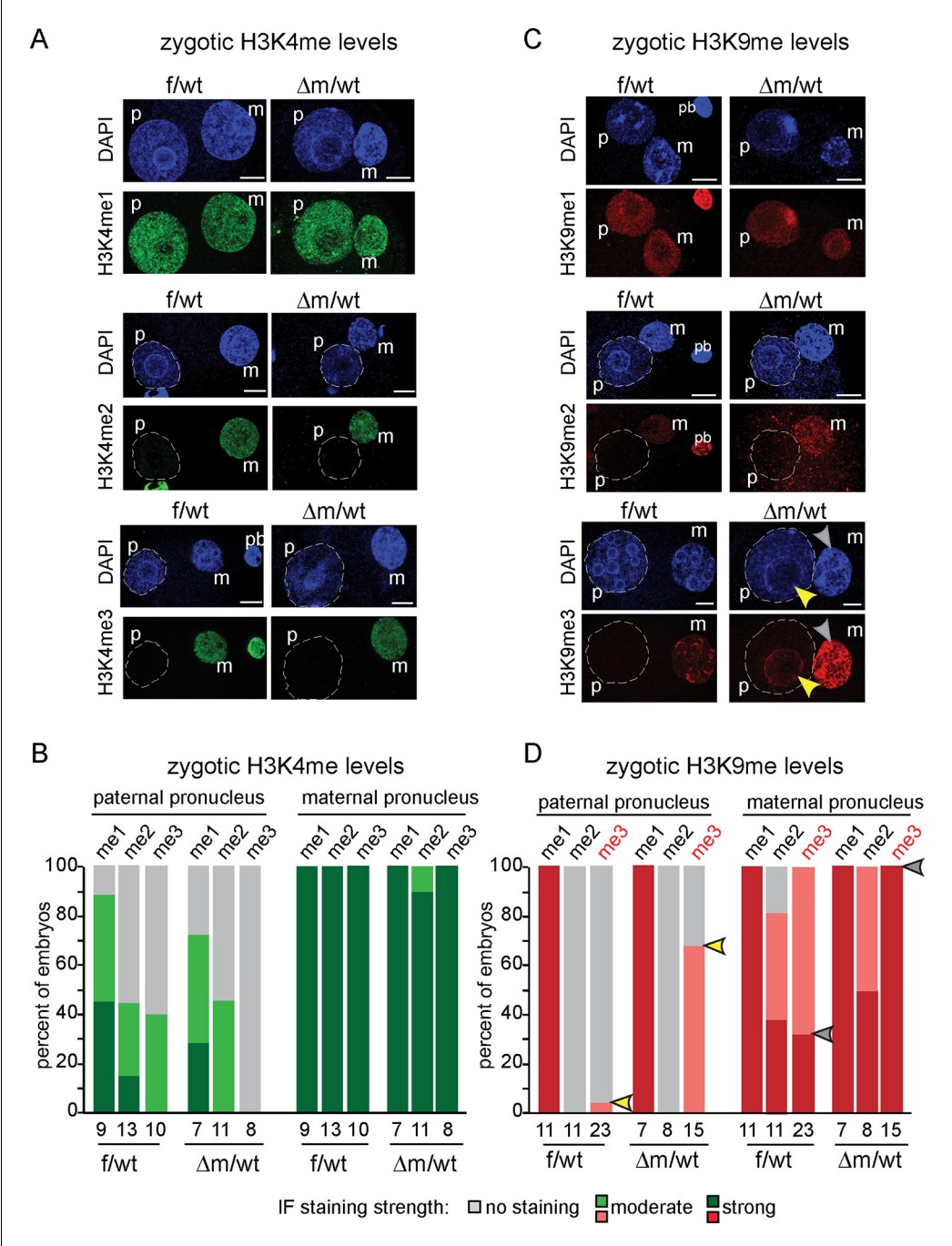

**Figure 2.** H3K9me3 heterochromatin levels are defined by maternally inherited KDM1A at the zygote stage. (A and C) IF using antibodies against me1, me2 and me3 of (A) H3K4 (in green) and (C) H3K9 (in red) during zygotic development. Mid to late f/wt and △m/wt zygote are shown. Paternal pronucleus (p), maternal pronucleus (m) and the polar body (pb) are indicated when present. DNA is counterstained with DAPI (blue). In C, note that in △m/wt zygotes, H3K9me3 is increased in the maternal pronucleus (grey arrowhead) and is localized *de novo* in the paternal pronucleus (yellow arrowhead). (B and D) Classification of embryos based on staining intensity scores for H3K4/K9me1/2/3 in the paternal versus maternal pronuclei in zygotes. Note that concerning H3K9me2, 50% of △m/wt embryos have a strong staining versus 35% in controls (which are also up to 20% with no IF signal). The most striking and only significant differences in proportions are seen for H3K9me3 both in maternal (grey arrowheads) and paternal (yellow arrowheads) pronuclei, with p<0.05 using a Chi square test. The scoring is as follows: light grey for no signal; medium green/red for moderate signal and dark green/red for strong signal. Number of embryos and their genotypes are indicated at the bottom of the graph. Scale bar in A and C represent 10 µm.

## Abnormal H3K4 and H3K9 methylation patterns after the first cleavage division in KDM1A maternally depleted embryos

In order to investigate whether KDM1A activity was important for the regulation of H3K4 and K9 methylation after the first cell cleavage, we examined two-cell stage f/wt and △m/wt embryos by IF to measure the relative fluorescence intensities (*Figure 3*). We found that the overall H3K4 methylation levels for mono, di and tri-methylation were significantly elevated in △m/wt two-cell embryos (*Figure 3A*), with the most striking effect being seen for H3K4me3 where a six-fold increase was found in mutants compared to controls. Thus, a lack of KDM1A protein has a significant impact on H3K4 methylation levels at the two-cell stage. When H3K9me1, me2 and me3 levels were also examined by IF, we found that all three marks were elevated, with the most significant effect being seen for H3K9me3, which showed a 2.2 fold increase in fluorescence intensity particularly at DAPI dense regions of constitutive heterochromatin (*Figure 3B*).

To address the specificity of these effects of KDM1A on H3K4 and H3K9 methylation, we tested other histone marks, reported not to be targeted by KDM1A activity. Two such marks, H3K27me3 and H4K20me3, both associated with heterochromatin, were analysed by IF in f/wt and △m/wt two-cell stage embryos. No significant changes in either of these marks could be detected in mutant compared to control embryos (*Figure 3—figure supplement 1A and B*), underlining the specificity of the defects found in KDM1A maternally depleted embryos. As an additional control, we performed IF analysis of two-cell stage embryos generated from wild-type zygotes grown for 24 hr with pargyline. H3K4me3 and H3K9me3 patterns revealed changes in pargyline-treated when compared to mock-treated embryos (*Figure 3—figure supplement 1C and D*). In both, a global increase in staining was detected when compared to controls, although to a slightly lesser extent than in *Kdm1a* mutant embryos.

## Absence of KDM1A abrogates the normal changes in transcriptome by the two-cell stage

After fertilization, development initially proceeds by relying on the maternally inherited pool of RNA and protein, followed by massive induction of transcription of the zygotic genome in different waves as shown in *Figure 4A*. Newly produced transcripts corresponding to zygotic genome activation (ZGA) appear in two phases, first at the zygote stage (corresponding to minor ZGA) and subsequently at the two-cell stage (major ZGA). Transition from the maternal pool to zygotic products is essential for successful developmental progression (*Flach et al., 1982*). Previous work has shown that KDM1A affects transcription regulation during *in vitro* embryonic stem cell (ESC) differentiation or during peri- or post-implantation mouse development (*Foster et al., 2010*; *Macfarlan et al., 2011*; *Wang et al., 2007*; *Zhu et al., 2014*). However, its role has never been evaluated during the very first steps of embryogenesis, when appropriate transcriptional activity is crucial.

In the light of our results on chromatin changes described above, and to assess whether transcription might be affected by the lack of the KDM1A maternal pool, we performed IF analysis against PolII and its elongating form (PolIISer2P), which did not reveal any obvious difference between f/wt and △m/wt two-cell stage embryos (*Figure 4—figure supplement 1A*). For a direct comprehensive analysis of the transcriptome upon lack of KDM1A, we used RNA sequencing (RNA-seq) for single oocytes and single embryos at the two-cell stage on a cohort of control and mutant samples (*Figure 4*; *Figure 4—figure supplement 2*; *Supplementary file 1*). The method used is based on that of (*Tang et al., 2010*) which captures poly(A) tail mRNA and allows examination up to 3kb from the 3' end. The quality of our single oocyte or embryo cDNAs was first checked by qPCR for three housekeeping genes (*Hprt, Gapdh, Ppia*) known to be stably expressed from oocytes to blastocysts (*Mamo et al., 2007*; *Vandesompele et al., 2002*). Control and mutant samples displayed similar relative expression for these three genes attesting to the quality of our samples (*Figure 4—figure supplement 1C*). We next prepared cDNA libraries from single control and mutant oocytes (n = 5 each), as well as individual f/wt and △m/wt embryos (n = 8 each) and performed Illumina-based deep RNA sequencing on these samples (see experimental procedures and analysis for more details; *Supplementary file 1*). We used DEseq as a normalization method across our samples to assess the relative gene expression between controls and mutants.

At the two-cell stage, our analysis revealed two sets of genes that become either upregulated (21%; n = 2449; FDR = 5%) or downregulated (24%; n = 2749) in the mutant when compared to

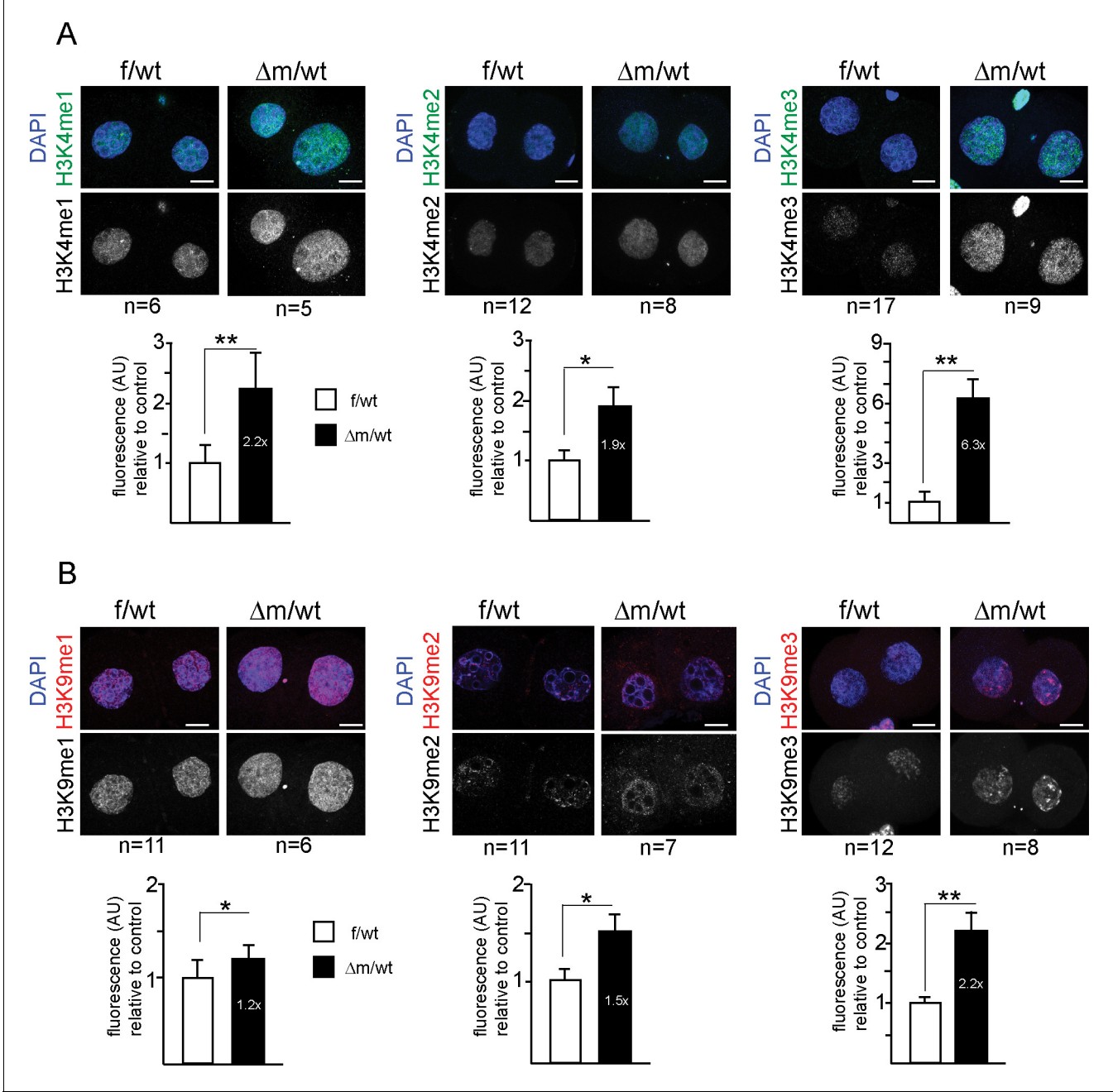

**Figure 3.** Two-cell stage H3K4 and H3K9 methylation levels are altered upon absence of maternal KDM1A. Immunofluorescence stainings of two-cell stage embryos using antibodies against me1, me2 and me3 of H3K4 (A; in green) and H3K9 (B; in red) were performed on f/wt (left panels) and △m/wt embryos (right panels). Control and mutant samples were processed in parallel and acquired using similar settings at the confocal microscope. DNA is counterstained with DAPI (blue). Projections of z-stacks are shown of representative embryos for each staining. Scale bars, 10 μm. Error bars represent S.E.M. By t-test; $p<0.05$ corresponds to * and $p<0.001$ to ** as performed on the number of embryos indicated below each picture. Below each image are shown the relative quantifications for IF intensity levels of me1, me2 and me3 of △m/wt (in black) relative to f/wt (in white) in two-cell stage embryos. Note that no alteration for H3K27me3 or H4K20me3 could be detected for mutant two-cell stage embryos (Figure 3—figure supplement 1A and B). Also, IF for pargyline-treated two-cell stage embryos revealed changes in both H3K4me3 and H3K9me3 patterns (Figure 3—figure supplement 1C and D).

The following figure supplement is available for figure 3:

**Figure supplement 1.** Immunofluorescence analysis of histone tail modifications upon maternal depletion or upon chemical inhibition of KDM1A for two-cell stage embryos.

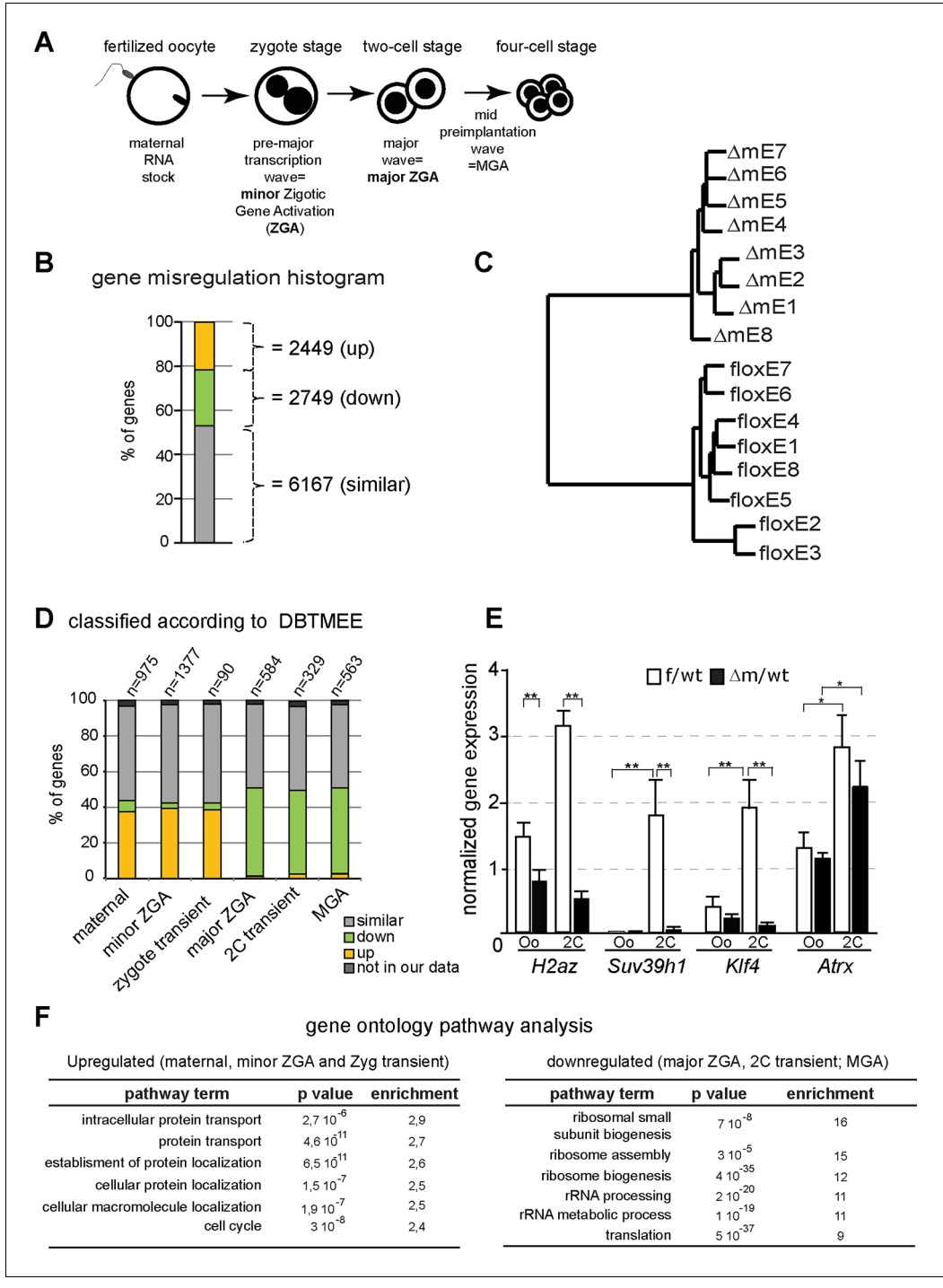

**Figure 4.** Abnormal ZGA upon absence of KDM1A revealed by transcriptome analysis. (**A**) Schematic illustration of the sequential sources of RNA pool over embryonic development. (**B**) Histogram shows the percent of differentially expressed genes in the △m/wt versus f/wt embryos. Fold difference (in log2) is annotated as upregulated (with log2≥ 1; yellow), downregulated (as log2≤-1; green) and similar (as 1<log2>-1; grey). Number of genes is indicated on the right of the graph. Details concerning the RNA seq analysis are described in Materials and Methods section and *Supplementary file 1* (**C**) Hierarchical clustering analysis for gene expression pattern of 16 libraries shows dramatic expression changes between f/wt (floxE1 to E8) and △m/wt (△mE1 to E8) two-cell stage embryos. See also *Figure 4—figure supplement 2* for analysis between two-cell stage and oocyte transcriptomes (**D**) RNA-seq data comparison with the different categories of the gene catalogue available at the Database of Transcriptome in Mouse Early Embryos (DBTMEE) generated an the ultralarge-scale transcriptome

*Figure 4 continued on next page*

*Figure 4 continued*

analysis (*Park et al., 2013*). The total number of genes belongings to each class and found in our RNA seq is indicated on top of the graph (see also *Table 1*). (**E**) Graphical representation of the normalized mean expression levels ± sem for chromatin-encoding genes in f/wt (in white) or △m/wt (in black) MII oocytes (Oo, n = 7) and two-cell stage embryos (2C, n = 10). *corresponds to p<0.05 and ** to p<0.001. (**F**) Top 6 representative GO terms (biological functions) enriched in △m/wt mutant embryos. Fold overrepresentation indicates the percentage of misregulated genes in a particular category over the percentage expected on the basis of all GO-annotated genes present within the sequencing. p-value indicates the significance of the enrichment.

The following figure supplements are available for figure 4:

**Figure supplement 1.** Immunostainings and RTqPCR analysis for assessing transcription of *Kdm1a* mutant two-cell stage embryos.

**Figure supplement 2.** Transcriptome analysis of *Kdm1a* mutant versus control in oocytes or two-cell stage embryos.

control embryos (*Figure 4B*; *Figure 4—figure supplement 2A*). Hierarchal clustering based on the transcription profiles showed that all *Kdm1a* mutant embryos clustered distinctly from the controls (*Figure 4C*). Furthermore, the analysis of oocyte transcriptomes also revealed that there were fewer genes misregulated in *Kdm1a* mutant oocytes, than in *Kdm1a* mutant embryos (*Figure 4—figure supplement 2A*). Moreover, Principal Component Analysis demonstrated that the gene expression patterns showed greater differences between controls and mutants at the two-cell stage, than in oocytes, and that the two different stages cluster away from each other (*Figure 4—figure supplement 2B*). The stage comparison also showed that only a subset of genes were misregulated in common, in both oocytes and two-cell stage embryos, upon loss of maternal KDM1A (*Figure 4—figure supplement 2C*). GO analysis of up or down regulated genes at the two-cell stage (*Figure 4F*) or oocytes (*Figure 4—figure supplement 2D*) revealed very little overlap in the specific biological functions affected by loss of function of KDM1A before and after fertilization, with the notable exception of cell cycle associated genes. This connects well with the observed phenotype for poor oocyte competence at fertilization and the total developmental arrest at the two-cell stage. These results reveal that absence of maternal KDM1A most likely leads to transcriptome changes during oocyte maturation, but to even more serious defects after zygotic gene activation, at the two-cell stage. The latter may be due in part to an aberrant maternal supply of transcripts/proteins, or else to aberrant transcriptional regulation of the zygotic genome in absence of maternal KDM1A.

We assessed our two-cell stage RNA-seq data according to the recent Database of Transcriptome in Mouse Early Embryos (DBTMEE) (*Park et al., 2013*). DBTMEE was built from an ultra-large-scale whole transcriptome profile analysis of preimplantation embryos, in which genes are classified depending on which transcription waves (as in *Figure 4A*) they are expressed. As shown in *Figure 4D* (see also Table 1), we assessed the percentage of genes of each of our classes (up; down and not significantly changed) that overlapped with the different DBTMEE categories of transcription switches, from oocyte to two-cell stage. Strikingly, the upregulated genes in △m/wt embryos fall essentially into the earliest stages and belong to genes annotated as maternal (37% of this category), as minor ZGA genes (39%) and as zygotic-transient (38%). We checked whether the misregulation of these three categories of genes might originate from the oocyte stage changes. We found that only 56 out of 360 of maternal genes, 94 out of 540 of minor ZGA genes and 6 out of 34 of 1C transient (Table 1 and data not shown) were already upregulated in mutant oocytes. These results reinforce the conclusion that the maternal and zygotic pools of transcripts become more compromised as development proceeds toward the two-cell stage in mutant embryos, rather than being aberrant right from the *Kdm1a* mutant germline. In clear contrast, the majority of downregulated genes in the △m/wt were found to belong to the three categories of genes that are normally activated at the two-cell stage, with 50% in the major ZGA class, 37% in the two-cell transient and 50% in the MGA (Mid zygotic gene activation). This suggests that absence of KDM1A compromises the activation of gene expression by the two-cell stage.

In order to validate our RNA seq data and the analysis done, we selected four genes with characteristic expression profiles, *Atrx* (maternal), *H2Az* (major ZGA), *Suv39h1* (2C-transient), *Klf4* (MGA),

which all encode chromatin associated factors crucial for early mouse development. Validation was performed by RT-qPCR in control and mutant oocytes and two-cell embryos. As predicted from our RNA seq results (*Supplementary file 2*), *H2AZ, Suv39h1 and Klf4* failed to be expressed at two-cell stage in *Kdm1a* mutant embryos (*Figure 4E*). In contrast *Atrx* which is a known maternal factor, but which is zygotically expressed by the two cell stage, was correctly activated. No difference in expression of *Suv39h1, Klf4* and *Atrx* could be seen between controls and mutants at the oocyte stage, implying that the maternal pool of these mRNAs was not affected by the maternal KDM1A depletion.

This single embryo transcriptome profiling data reveals an aberrant gene expression profile in *Kdm1a* mutant embryos, which is likely due to an absence or delay in the transcription switch from maternal-zygote to the two-cell stage pattern for a substantial set of genes (47%; 1818 out of 3811 considered; *Figure 4—figure supplement 2*). Together with the changes in chromatin profiles that we observed at the two-cell stage, we conclude that part of the deficiency in developmental progression could be due to the inappropriate setting of a successful zygotic gene expression program upon KDM1A loss.

A gene ontology (GO) analysis of the up-regulated genes classified as maternal to zygote-transient in *Figure 4D*, revealed a clear over-representation of genes involved in protein transport and localisation as well as contribution to cell cycle (*Figure 4F*). GO analysis of the downregulated genes from major-to-mid-zygotic activation are implicated in ribosome biogenesis and translation processes (*Figure 4F*). Collectively, these results suggest that KDM1A is necessary for the transcriptional regulation of specific genetic pathways implicated in fundamental biological functions such as protein production and localisation, and cell cycle regulation. These combined defects could be consistent with the inability of the mutant embryos to develop further than the two-cell stage

## Impact of KDM1A absence on repeat elements, genome integrity and DNA replication

Many transposable elements are known to be expressed in early mouse embryos, as early as zygotic stage, and some of these repeat elements might even be competent for new events of retrotransposition between fertilization and implantation (*Fadloun et al., 2013*; *Kano et al., 2009*; *Peaston et al., 2004*). The repression of some of these transposable elements during preimplantation has been correlated with loss of active chromatin marks such as H3K4me3, rather than acquisition of heterochromatic marks such as H3K9me3 (*Fadloun et al., 2013*). Interestingly, a previous study using *Kdm1a* mutant mESCs and late preimplantation embryos found a significant impact on MERVL:LTR repeat expression (for Murine endogenous retrovirus-like LTR), as well as a good correlation for the presence of remnant ERVs within 2kb of the transcription start site of KDM1A-repressed genes (*Macfarlan et al., 2011*). The increased levels of H3K4me3 and H3K9me3 that we found in △m/wt two-cell stage embryos and the reported role of KDM1A in late preimplantation embryos prompted us to analyze the effects of maternal KDM1A depletion on repetitive element expression after fertilization. To this end, we investigated our RNA-seq data from control and *Kdm1a* mutant

**Table 1.** Comparing the two-cell stage transcriptome of the *Kdm1a* mutant embryos to DBTMEE Numbers of genes found for the comparison of our two-cell stage RNA-seq data with the different categories for the gene catalogue found in DBTMEE. Total genes considered = 3811 and total genes changed = 1818 (48%). Our dataset cover the genes categorized on the public resource with a minimum of 96% of genes.

|  | Up | Down | Similar | Not in our data | Total in DBTMEE |
|---|---|---|---|---|---|
| maternal | 360 | 63 | 521 | 31 | 975 |
| minor ZGA | 540 | 47 | 750 | 40 | 1377 |
| 1C transient | 34 | 4 | 51 | 1 | 90 |
| major ZGA | 10 | 297 | 297 | 10 | 584 |
| 2C transient | 8 | 156 | 156 | 12 | 329 |
| MGA | 13 | 286 | 286 | 13 | 563 |

two-cell embryos for the relative expression of repetitive elements. As our single embryo RNA seq approach was based on oligo-dT priming this restricted our analysis to reads at the 3' ends of transcripts, which somewhat limited our capacity to detect repeat variation. In particular we could not determine which specific LINE-1 families were expressed in the mutants, nor whether the LINE-1 reads we detected corresponded to full-length,and/or intact elements. Nevertheless, our results shows that by far the most abundant categories of expressed repeats at this stage of development were LTRs (long terminal repeat) and non-LTR retrotransposons in f/wt and △m/wt (95% and 92%, respectively) (*Figure 5A*). However, no significant impact on expression could be detected in the mutants, with the exception within the non-LTR elements, of quite a significant overrepresentation of LINEs, but not SINEs (for Long/Short Interspersed Nuclear Elements element) (*Figure 5A and B*). We validated this result by RT-qPCR using individually prepared cDNAs of two-cell stage embryos for three transposable element classes. LINE-1, SINE B1 and MuERV-L transcripts are all abundantly expressed in control and mutant embryos, but LINE-1 levels show a two-fold increase in the △m/wt embryos (*Figure 5C*). No significant up-regulation was seen in ERV-promoter driven genes, that had previously reported to be affected by loss of KDM1A in ESCs (*Macfarlan et al., 2011*).

To further assess the impact that KDM1A depletion has on active LINE-1 transcription, we used a single-cell method, RNA fluorescent in situ hybridization (RNA FISH), which enables the detection of nascent transcripts. We first assessed the quality of our assay by checking the *Atrx* gene, known to be transcribed at the two-cell stage (*Patrat et al., 2009*) and expressed at similar levels in mutant and control (*Figure 5—figure supplement 1*). A comparable proportion of f/wt embryos and △m/wt two-cell embryos displayed detectable ongoing transcription, as registered by a pinpoint at this locus. Using a probe spanning the full-length LINE-1 element (*Chow et al., 2010*), we detected LINE-1 RNA in control two-cell stage embryos as displayed by the punctate pattern in nuclei (*Fadloun et al., 2013*), while RNase-A treated embryos showed no signal (*Figure 5D*). In the maternally depleted embryos, the arrangement of fluorescent foci appeared extensively modified (*Figure 5D*). This was confirmed upon analysis of the fluorescence intensity distributions (*Figure 5E* left) as well as the image composition for the foci (*Figure 5E* right), which in both cases significantly separated the two types of samples. Our analysis revealed that the active LINE-1 transcription profiles were extensively modified upon the loss of maternal KDM1A.

To investigate whether the increase in nascent LINE-1 transcription observed might correspond to full length LINE-1 elements, we assessed by IF for the presence of ORF1, one of the two LINE-1 encoded proteins. At the two-cell stage, we found an approximately four-fold increase in the proportion of △m/wt embryos displaying a stronger IF signal, notably in the nucleus (*Figure 5F*). These results suggest that the LINE-1 deregulation observed at the RNA level might indeed lead to the production and nuclear import of increased levels of LINE-1 ORF1 proteins. We next investigated whether expression of such proteins from transposable elements would have any consequences. We thus performed γH2AX IF staining to assess whether increased DNA damage signalling could be seen in △m/wt compared to f/wt embryos (*Figure 5G*; *Figure 5—figure supplement 2*). Half of the mutant embryos displayed a stronger staining for γH2AX, with a significant increase compared to controls (*Figure 5G*). We also assessed whether this accumulation of γH2AX signals could also be related to replication delays, as reported previously in the case of maternal loss of two components of the polycomb complex PRC1 (*Posfai et al., 2012*). We performed EdU pulse treatment (a nucleoside analog of thymidine incorporated into DNA) in two-cell embryos, at a stage when they have normally completed S phase (40–41 hr post hCG injection). This revealed that S phase is delayed in the △m/wt embryos given the incorporation of EdU in the mutants, while none of the control embryos used in parallel were stained (*Figure 5—figure supplement 2*). All the mutant embryos delayed in their replication displayed concomitantly intense γH2AX signals. However, 38% of the mutant embryos did not show any EdU incorporation, indicating that they exit S phase, yet, they still show high levels of γH2AX signals. Finally, although, no significant enrichment was directly found for DNA damage pathways when running our GO analysis (*Figure 4E*), many genes related to DNA damage repair were upregulated (*Supplementary file 2*). Taken all together, these results suggest that the elevated DNA damage signalling observed could be independent from replication defaults in KDM1A maternally depleted embryos, but might be related either to changes in transcript levels for DNA damage genes or else to the observed increase in LINE-1 activity in *Kdm1a* mutant embryos at this stage.

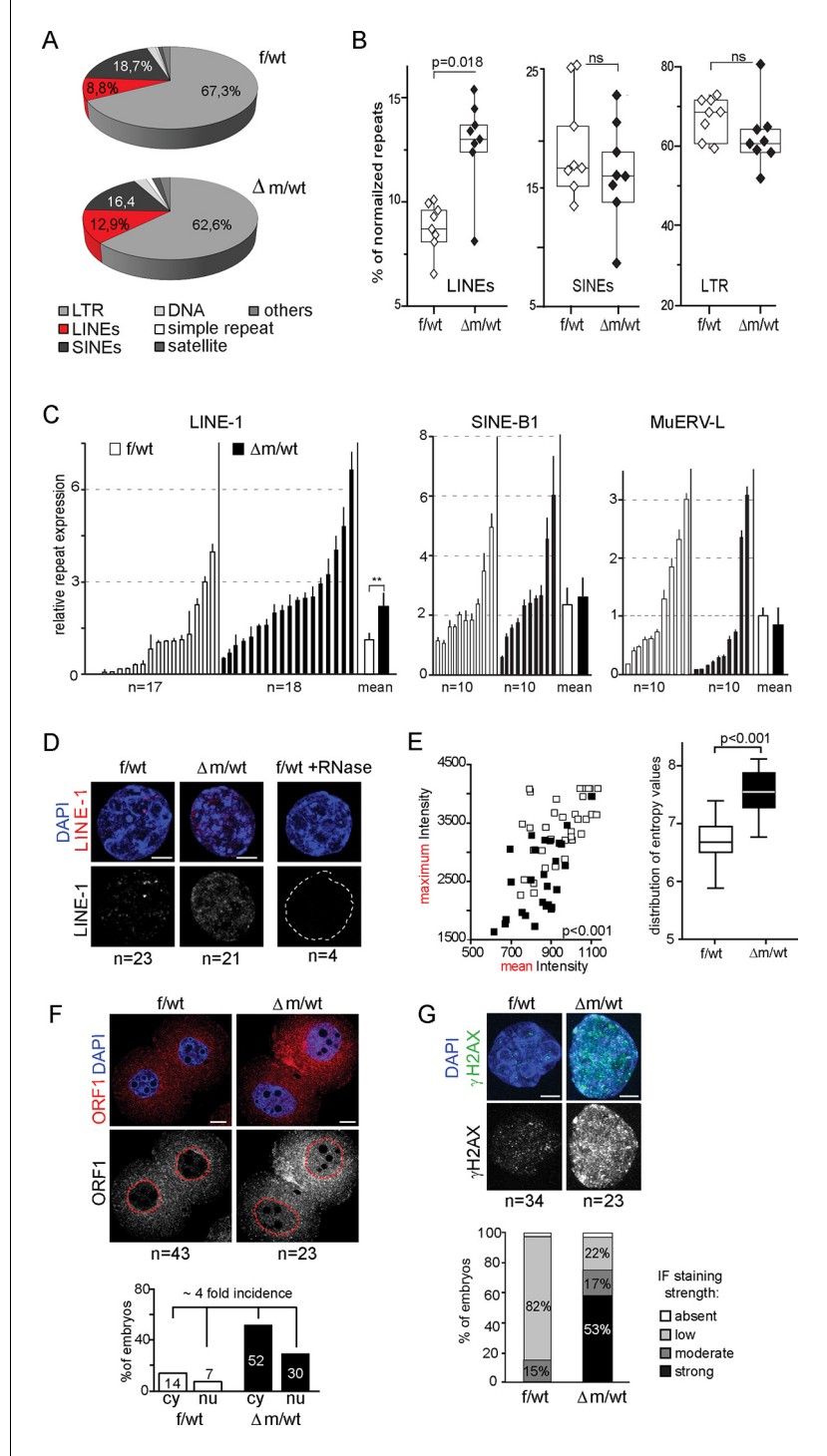

**Figure 5.** Increased LINE-1 protein levels and γH2AX foci in two-cell embryos depleted for KDM1A. (**A**) Pie chart representing the percent of each category of repeats analyzed in our 16 RNA-seq data of individual embryos. (**B**) Box-plot for percent of LINEs, SINEs and LTR element expression for f/wt (white) or △m/wt (black) embryos over the total of reads mapping repeats for each of our 16 samples of RNA-seq. Details of the analysis are in experimental analysis. (**C**) qPCR analysis for LINE-1, SinesB1 and MuERV-L expression levels from individual two-cell stage cDNAs of f/wt (white) or △m/wt (black). Each embryo is represented as a single bar. Data are expressed as normalized expression to three house-keeping genes. On the right of each graph is represented the mean ± sem. Two asterisks indicate p<0.01 as calculated using a Student's t-test. (**D**) Nascent *LINE-1* transcripts are detected by RNA FISH (signal in red) using a TCN7 probe on f/wt or △m/wt two-cell stage embryos. RNAse A

*Figure 5 continued on next page*

*Figure 5 continued*

treated control embryos processed in parallel display no signal for RNA transcription. Also see *Figure 5—figure supplement 1* for *Atrx* expression control. (**E**) Quantification of LINE-1 RNA FISH. On the left, the graph represents the fluorescent quantification with the mean intensity of fluorescence plotted on the x axis against the respective maximum intensity on the y axis) for each nucleus of the two–cell embryos for the two populations (white squares = f/wt controls and black square = △m/wt mutants). On the right, box-plot representation of the entropy levels analysis of the RNA FISH images for the control versus mutant embryos as defined by Haralick parameters measuring the pattern of the image with each dot corresponding to a f/wt (white) or △m/wt (black) nucleus.P value is calculated with a student T-test and indicated that the two populations are significantly different. (**F**) IF of two-cell stage embryos using anti-ORF1 antibodies (in red). A dotted line indicates the nucleus. Below is the graphical representation of the percentage of embryos displaying enriched fluorescent signal in either the cytoplasm (cy) or the nucleus (nu) for f/wt or △m/wt embryos. (**G**) IF of two-cell stage embryos using antibodies directed against phosphorylated histone H2A variant X (γH2AX, in green) for f/wt and △m/wt. Below is the corresponding quantification of embryo percentage according to the strength of γH2AX staining. DNA is counterstained by DAPI (blue). Number of processed embryos is indicated. Scale bar, 2 μm (**D**, **G**)) 10 μm (**F**).

The following figure supplements are available for figure 5:

**Figure supplement 1.** RNA FISH controls for LINE-1 ongoing transcription.

**Figure supplement 2.** EdU labeling and γH2AX immunofluorescence of *Kdm1a* mutant versus control two-cell stage embryos.

## Discussion

The oocyte stores maternal factors that besides ensuring the first steps of development prior to zygotic genome activation, also enable the epigenetic reprogramming of the parental genomes (*Burton and Torres-Padilla, 2010*; *Li et al., 2010*; *Messerschmidt et al., 2012*; *Lorthongpanich et al., 2013*; *Seisenberger et al., 2013*). Although the dynamics of histone modifications have been assessed, the biological relevance of such changes and the identification of the histone modifying enzymes involved in this process are only starting to be identified. In this study, we have focused on the critical function of KDM1A, a histone demethylase for H3K4me1/2 and H3K9me1/me2, that we find acts as a maternal chromatin factor at the time egg fertilization. We show that maternal KO results in abnormal oocytes at the time of ovulation (at meiosis II stage) and prevents development of fertilized eggs beyond the two-cell stage. Similar oocyte defects and developmental block were also observed in the accompanying paper by Wasson et al, where two other independent conditional alleles were used to induce deletion of KDM1A in oocytes, using Zp3-Cre or Gdf9 Cre. In a recent study maternal depletion of KDM1A was found to affect the first division of meiosis and leads to early apoptosis during oocyte growth (*Kim et al., 2015*). Taken together, these studies show that KDM1A is required during the formation of the female gametes for the two steps of meiosis (*Kim et al., 2015*; this study; Wasson et al. ). Our study also reveals that KDM1A is required as a key factor during early post-zygotic embryo development, as the enzymatic inhibition of KDM1A in wild-type embryos resulted in developmental arrest at the two-cell stage, comparable to maternal deletion. We further show that KDM1A is a major regulator of histone H3K4 and H3K9 methylation patterns at the one-two cell stages and that it controls early switches in transcription patterns during development. KDM1A may also have a potential role in the appropriate repression of some LINE-1 retroviral elements.

### KDM1A and the modulation of histone methylation after fertilization

H3K4me1/2/3 levels have been shown to increase from the zygote to the two-cell stage, before decreasing again by the four-cell stage (*Shao et al., 2014*). To date, only one H3K4 KMT, MLL2, has been shown to be necessary at the two-cell stage (*Andreu-Vieyra et al., 2010*). Here, we report that the maternal pool of the KDM, KDM1A, is also necessary at this stage, with its loss leading to global elevation of H3K4me1/2/3. Noticeably, no changes in transcription levels of genes encoding the main H3K4me2/3 KMTs were recorded (*Supplementary file 2*). This suggests that KDM1A is a key regulator of H3K4 methylation post-fertilization.

Moreover, in absence of KDM1A, the transcripts encoding for two main KMTs (SUV39H2/KMT1B and SETDB1/KMT1E) targeting H3K9me1/2 during preimplantation (*Cho et al., 2012*; *Puschendorf et al., 2008*) are well detected in two-cell stage mutant embryos (*Supplementary file 2*). These KMTs could be able to generate H3K9me3 from the excess of H3K9me1/2, produced because of the absence of KDM1A.

In conclusion, KDM1A most likely acts in combination with other chromatin regulators in order to keep a tight balance of the global H3K4/K9 methylation levels during early embryonic development.

## KDM1A is involved in the transcriptional switch at the two-cell stage

One of the most striking consequences of lack of maternal KDM1A that we observed was the disruption of the wave-like gene expression patterns previously described at the onset of mouse development (*Hamatani et al., 2004*; *Xue et al., 2013*). At the two-cell stage, we saw a significant increase in mRNA levels of genes normally expressed maternally or at the zygote stage, and this increase relates more to post-fertilization disruption rather than inherited defects from *Kdm1a* mutant germline. The accompanying manuscript by Wasson et al reports similar findings concerning transcriptional regulation by maternal KDM1A in early stage post fertilization. Maternal and zygotic mRNA excess could reflect a reduced rate of mRNA degradation, maybe related to the developmental arrest, or else the severe impairment of the mutant embryos in the ribosome biogenesis pathways could preclude the translation machinery of their usage and clearance or else a change in the cytoplasmic polyadenylation of the maternal pool of mRNA could also be disturbing their utilization. Lastly, their abundance could also be due, maybe partly, to an increased transcription rate for these genes (and more specifically the one corresponding to the minor ZGA). Given the accumulation of H3K4 methylation that we show in our study for the mutants at this stage and the proven link of this mark with enhanced transcription (*Black et al., 2012*), we hypothesise that KDM1A might normally be involved in the transcriptional down-regulation of these genes via H3K4 demethylation. Chromatin-based repression is thought to be superimposed on zygotic genome activation and is necessary for the transition from the two-cell to the four-cell stage (*Ma, 2001*; *Ma and Schultz, 2008*; *Nothias et al., 1995*; *Wiekowski et al., 1997*). We propose that KDM1A might be part of such a mechanism, and required for a transition towards two-cell stage specific gene expression patterns (ie in the major ZGA and MGA waves), and therefore for proper development beyond the two-cell stage. The significant absence of the major ZGA and MGA waves in the transcriptome of *Kdm1a* mutants supports this hypothesis. Whether misplaced or increased H3K9 methylation (*Figure 3B*) could be involved in failure of transcription activation is not known, but one can speculate that such repressive chromatin and/or absence of KDM1A itself might impair correct recruitment of transcription regulators. So far, a small subset of such factors (TFs and co-regulators) acting at ZGA-gene promoters has recently been suggested to orchestrate the appropriate gene expression patterns following fertilization (*Park et al., 2013*; *Xue et al., 2013*). Although, KDM1A was not reported in this study, our results suggest that maternal KDM1A is nonetheless crucial for shaping the transcriptome in early life. Its role in oocyte and embryogenesis may have long lasting effects, as reported in the accompanying paper by Wasson et al where a hypomorphic maternal KDM1A, associated with perinatal lethality, showed alterations in imprinted gene expression much later in life. The importance of the maternal pool of KDM1A opens up exciting prospects for the roles of this remarkable histone demethylase in early development.

## KDM1A is instrumental in preserving the genome integrity

The control of repeat elements by epigenetic mechanisms, including histone KMTs and KDMs, may be critical in early development. Previous work has suggested that KDM1A may contribute to MERVL element repression in late pre-implantation embryos (*Macfarlan et al., 2011*). We did not detect any impact on these elements in the *Kdm1a* mutants immediately post fertilization. However, we did see a small but significant increase in LINE-1 expression and LINE-1 ORF1 protein levels in the *Kdm1a* mutant embryos. This observation, together with the striking elevation in H3K4me3 levels, is of particular interest in the context of a recent study which proposed that loss of H3K4m3 at LINE-1 elements (rather than a gain in H3K9 methylation) might be critical for their repression during early pre-implantation development (*Fadloun et al., 2013*). Whether this increase in LINE-1 expression actually leads to an increase in LINE-1 element retrotransposition (ie new insertions) remains to

be seen, but the increase in LINE-1 proteins observed in *Kdm1a* mutant embryos is potentially consistent with such a possibility. In this context, we speculate that misregulation of LINE-1 elements in the absence of KDM1A might participate in the early developmental arrest that is observed, via an increased potential of genome instability and activation of some specific DNA damage checkpoints. The increase in γH2AX foci we detected in *Kdm1a* mutants, independently from replication stalling problems, could also be consistent with this hypothesis. Our results thus support the hypothesis that histone-based defence mechanisms act to safeguard the genome from LINE-1 retrotransposition during preimplantation development, when global DNA hypomethylation might compromises their usual silencing route (*Leung & Lorincz, 2012*).

Finally, chromatin status and regulated expression of another family of repeats, located within pericentric heterochromatin, has been proposed to be involved in developmental progression after fertilization, ensuring correct chromosome segregation and heterochromatin propagation (*Probst et al., 2010*; *Santenard et al., 2010*). In the absence of KDM1A, we detected aberrant accumulation of H3K9me3 at presumptive pericentric heterochromatin (NLBs) post-fertilization, as well as lagging chromosomes in oocytes, and micronuclei accumulation following fertilisation. Collectively, this data points to maternal KDM1A protein having a potential role at pericentromere/centromere regions that merits future exploration.

In conclusion, our findings demonstrate the instrumental role of KDM1A as a maternally provided protein at the beginning of life in shaping the histone methylation landscape and the transcriptional repertoire of the early embryo.

## Materials and methods

### Experimental methods

#### Collection of mouse embryos and in vitro culture

All mice used were handled with care and according to the guidelines from French legislation and institutional policies. Mice (*Kdm1a^tm1Schüle^*) carrying the targeted mutation allowing the conditional deletion of the first exon of *Kdm1a* by insertion of two flanking LoxP sites has been engineered and described by R.Schüle group (*Zhu et al., 2014*). We received mice carrying two copies of this new conditional allele *Kdm1a^tm1Schüle^*, and after transfer in our animal facitilities, they were bred over the well know Zp3^cre^ deleter strain which allow CRE mediated recombination specifically in the female germline (*Lewandoski et al., 1997*). The genetic background of the mice *Kdm1a^f/f^::Zp3^cre^* is a mixture of C57BL/6J and a 129 substrains, and are referred in this manuscript as *Kdm1a^f/f^::Zp3^cre^* mice (as carrying two *Kdm1a* conditional alleles and a *Zp3.cre* transgene). To evaluate KDM1A functions during early development, embryos were obtained from superovulated *Kdm1a^f/f^::Zp3^cre^* or *Kdm1a^f/f^* females (aged 4–8 weeks) mated with B6D2F1 males (see *Figure 1C*), and collected in M2 medium (Sigma, Saint-Louis, MO) at 21–28 hr (zygote) and 40–42 hr (two-cell) after hCG (human chorionic gonadotropin) injection. For pargyline treatment (Sigma;1 mM final during 24 hr) zygotes were in vitro cultured in M16 (Sigma) droplets under mineral oil in a 5% CO2 atmosphere at 37°C. For replication assays, two-cell-stage embryos were collected at 39–40 hphCG, and embryos were cultured in M16 medium 1 hr, then transferred to M16 containing 50 µM EdU (Click it Life Technologies, Santa Clara, CA) for 45 min. Following fixation in 4% PFA for 15 min, permeabilization in PBS 0.5% Triton X-100 for 15 min, blocking in PBS 3% BSA. Click-it reaction was performed for 1 hr. Washes and new blocking were followed by immunostaining with antibodies against γH2AX (see next section).

### Immunofluorescence staining

Immunofluorescence was carried out as described previously (*Torres-Padilla et al., 2006*), with some modifications. After removal of the zona pellucida with acid Tyrode's solution (Sigma), embryos were fixed in 4% paraformaldehyde, 0,2% sucrose, 0.04% Triton-X100 and 0.3% Tween20 in PBS for 15 min at 37°C. After permeabilisation with 0.5% Triton-X100 in PBS for 20 min at room temperature, embryos were washed in PBStp (0.05% Triton-X100; 1 mg/ml polyvinyl pyrrolidone (PVP;Sigma)) then blocked and incubated with the primary antibodies in 1% BSA, 0.05% Triton-X100 for ~16 hr at 4°C. Embryos were washed in PBStp twice and blocked 30 min in 1% BSA in PBStp and incubated for 2 hr with the corresponding secondary antibodies at room temperature. After

washing, embryos were mounted in Vectashield (Vector Laboratories, Burlingame, CA) containing DAPI (4',6'-diamidino-2-phénylindole) for visualizing the DNA. Full projections of images taken every 0.5 μm along the z axis are shown for all stainings, except for the ORF1 for which the middle section is shown only. Antibody staining for H3K4 methylation is in green, and in red for H3K9 methylation, DNA is counterstained with DAPI (blue). For each antibody, embryos were processed identically and analyzed using the same settings for confocal acquisition Stainings were repeated independently at least twice. The following antibodies were used (Antibody/Vendor/Catalog #/Concentration): anti-rabbit KDM1A/Abcam (UK)/ab17721/ 1:750, anti mouse H3K4me1/Cosmobio (Japan)/MCA-MBAI0002/ 1:700, anti mouse H3K4me2/Cosmobio /MCA-MBAI0003/ 1:700, anti mouse H3K4me3/ Cosmobio/MCA-MBAI0004/ 1:700, anti-rabbit H3K9me1 kind gift from T.Jenuwein, anti mouse H3K9me2/Cosmobio /MCA-MBAI0007/ 1:500, anti rabbit H3K9me2/ActiveMotif (Carlsbad, CA) / 39239/ 1:800, anti rabbit H3K9me3/ Millipore (Billerica, MA/07–442/ 1:200, anti-mouse H3K27me3/ Abcam/ab6002/ 1:400, anti-rabbit H4K20me3/ Abcam/ab 9053/ 1:200, anti-mouse γH2AX/ Millipore/05–623/ 1/200, anti-mouse β-TUBULIN/ Invitrogen (Carlsbad, CA)/32–2600/ 1:1000, anti-mouse POLII CTD4/ Millipore/05–623/1:200, anti-rabbit POLII CTD4 S2P/Abcam/ab5095/1:200, anti-rabbit ORF1, kind gift from A.Bortvin/ 1:500, Alexa488 goat anti-mouse IgG/ Invitrogen/A11029/ 1:500, Alexa568 goat anti-rabbit IgG/ Invitrogen A11036/ 1:500.

## Western-blot procedure

50 two-cell stage embryos were resuspended in 2-mercaptoethanol containing loading buffer and heated at 85°C for 15 m. SDS-PAGE, Ponceau staining, and immunoblots were performed following standard procedures using a Mini-PROTEAN Tetra Cell System (Bio-Rad, Hercules, CA). Primary anti-KDM1A (dilution 1:500) and secondary HRP-conjugated goat anti-rabbit (DAKO, Santa Clara, CA, Cat.#K4002) were used. 2 μg of ESC nuclear extracts were used as control.

## RNA FISH procedure

RNA FISH was performed as described (*Patrat et al., 2009*). Nick translation (Vysis Abbott, Chicago, IL) using Spectrum green or Spectrum red (Vysis) was used to label double stranded probes. The LINE-1 probe used consisted of a full- length Tf element cloned into a Bluescript plasmid as previously described (*Chow et al., 2010*). The *Atrx* probe consisted of a BAC (CHORI, Oakland, CA; reference RP23-260I15). Briefly, embryos were taken at 42 hr post hCG and the *zona pellucida* was removed. Embryos were transferred onto coverslips previously coated in Denhardt's solution, dried down for 30 min at room temperature, after all excess liquid was removed. Samples were fixed in 3% paraformaldehyde (pH 7.2) for 10 min at RT and permeabilized in ice-cold PBS 0.5% triton for 1 min on ice and then directly stored in ETOH 70°C ethanol at -20°C until processed for RNA FISH. Hybridizations, without *Cot1* competition for LINE-1, were performed overnight at 37°C in a humid chamber. Excess of probes was eliminated through three washes in 2xSSC at 42°C for 5 min each. Slides were mounted in Vectashield containing DAPI.

## Single embryo RNA RT-qPCR and deep sequencing

After *zona pellucida* removal and 3 consecutive washes in PBS-0.1% BSA, individual oocytes or whole two-cell stage embryos were transferred into a 0.2 ml eppendorf tube (care was taken to add a minimum liquid volume of PBS BSA) and directly frozen in -80°C until use. RNA was extracted and amplified as described previously (*Tang et al. 2010*). For quality control and gene expression analysis, quantitative real-time PCR was performed for gene expression on 1/10 dilution of cDNA preparation in 10 μl final volume with Power SYBR green PCR master mix (Applied Biosystems, Foster City, CA) on a ViiA7 apparatus (Life Technologies). The level of gene expression was normalized to the geometric mean of the expression level ofFoster City Hprt, Gapdh and Ppia housekeeping genes as according to (*Vandesompele et al., 2002*). For p<0.05 corresponds to * and p<0.001 to ** by t-test. The following primers used in this study are listed as name/ forward primer 5' to 3' / reverse primer 5' to 3' *Hprt*/ ctgtggccatctgcctagt / gggacgcagcaactgacatt, *Gapdh*/ ccccaacactgagcatctcc / attatgggggtctgggatgg, *Ppia*/ ttacccatcaaaccattccttctg / aacccaaagaacttcagt-gagagc (as in *Duffie et al., 2014*Atrx/ tgcctgctaaattctccaca / aggcaagtcttcacagctgt, *H2AZ*/ acacatc-cacaaatcgctga / aagcctccaacttgctcaaa, *Klf4*/ agccattattgtgtcggagga/ agtatgcagcagttggagaac, *Suv39h1*/ ctgggtccacttgtctcagt/ ctgggaagtatgggcaggaa, SineB1/ gtggcgcacgcctttaatc /

gacagggtttctctgtgtag (*Martens et al., 2005*), MuERVL/ atctcctggcacctggtatg / agaagaaggcatttgc-caga (*Macfarlan et al., 2011*), *Kdm1a*/ tggagaacacacaatccgga / tgccgttggatctctctgtt, LINE-1 3'UTR/ atggaccatgtagagactgcca / caatggtgtcagcgtttgga

For RNA deep sequencing, library construction was performed following Illumina (San Diego, CA) manufacturer suggestions. The 26 samples (5 f/f or wt/wt and 5 Δm/Δm oocytes; 8 f/wt and 8 Δm/wttwo-cell embryos) were sequenced in single-end 49 bp reads on an Illumina HiSeq 2500 instrument. The depth of sequencing was ranged from 12,500,000 to 35,000,000 with an average around 18,000,000 reads per sample (*Supplementary file 1*).

## Data procession and analysis
### Confocal acquisition and image analysis
Imaging of embryos following IF and FISH was performed on an inverted confocal microscope Zeiss (Germany) LSM700 with a Plan apo DICII (numerical aperture 1.4) 63x oil objective. Z sections were taken every 0.4 μm (*Figure 1–3*) or 1 μm (*Figure 4* and *5*). For fluorescence intensity measurement on immunofluorescence Z stacks acquisitions, the nuclear area of the stack image was selected, and then the integrated Intensity (intensity divided by the number of voxels represented within the nuclear area) was obtained using the 3D object counter plugin in Image J (*Bolte and Cordelieres, 2006*). For LINE-1 RNA FISH analysis, home-made script for ImageJ were developed that used descriptors defined as (*Haralick, 1979*) to quantitatively study the texture and structure of images (see related manuscript file containing the code in Java text). Distribution of fluorescence intensities or of Haralick parameters (eg entropy) were compared using t-tests, after all data had been tested as belonging to normally distributed populations (Origin8Pro software, Northampton, MA). For $p<0.05$ corresponds to * and $p<0.001$ to **.

## RNA sequencing
For the gene-based differential analysis, quality control was applied on raw data. Sequencing reads characterized by at least one of the following criteria were discarded from the analysis: (more than 50% of low quality bases (Phred score <5); more than 5% of N bases; more than 80% of AT rate At least 30% (15 bases) of continuous A and/or T). Reads passing these filters were then aligned to the mouse mm10 genome using the TopHat software v2.0.6 (*Trapnell et al., 2009*). Only unique best alignments with less than 2 mismatches were reported for downstream analyses. Count tables of gene expression were generated using the RefSeq annotation and the HTSeq v0.6.1 software (*Anders et al., 2015*). The DESeq R package v1.16.0 (*Anders and Huber, 2010*) was then used to normalize and identify the differentially expressed genes between control and mutant embryos. Genes with 0 counts in all samples were filtered out and only the 60% of the top expressed genes were used for the analysis, as described in the DESeq reference manual. Genes with an adjusted p-value lower than $\alpha = 0.05$ were consider as differentially expressed. Hierarchical clustering analysis for gene expression pattern of 16 libraries was based on Spearman correlation distance and the Ward method, and performed using the hclust function implemented in the gplots v2.16.0 R package.

In order to study the transposons expression, we performed the mapping of reads passing the quality control using the Bowtie v1.0.0 software (*Langmead et al., 2009*). This mapping was performed in 2 steps: (i) reads aligned on ribosomal RNA (unique best alignments with less than 3 mismatches in the seed) (GenBank identifiers:18S, NR_003278.3; 28S, NR_003279.1; 5S, D14832.1; and 5.8S, KO1367.1) were discarded (ii) remaining reads were aligned to the mouse mm10 genome, reporting a maximum of 10,000 genomic locations (best alignments without mismatches). Aligned reads were then annotated and intersected with repeats annotation from the repeatMasker database. The transposon counts table was generated using the reads that fully overlap with an annotated repeat and for which all possible alignments are concordant, i.e associated with the same repeat family in more than 95% of cases. The resulting count table was normalized by the total number of reads aligned on repeats. Statistical analysis to identify repeat families with significant changes in expression between control and mutant embryos was performed using the limma R package v3.20.4 (*Ritchie et al., 2015*). Repeats family with an adjusted p-value lower than $\alpha = 0.05$ were consider as significant.

The tool AmiGO 2 (*Carbon et al., 2009*) was used to perform the enrichment Gene Ontology items with the misregulated genes from the *Kdm1a* mutant two-cell stage embryos.

## Data access

The Gene Expression Omnibus (GEO) accession number for the data sets reported in this paper is GSE68139

## Acknowledgements

We thank A Bortvin and T Jenuwein for kind gift of antibodies. We thank Simao Teixeira Da Rocha, Rafael Galupa and Petra Hajkova for critical reading of the manuscript, and members of the EH laboratory for feedback. We acknowledge the pathogen-free barrier animal facility of the Institut Curie, in particular Colin Jouhanneau, and the UMR3215/U934 Imaging Platform (PICT-IBiSA), in particular Olivier Leroy and Nicolas Signolle. MB was supported by the DIM-Stem Pôle, Ile de France, Funding for EH: Equipe labellisée ''La Ligue Contre Le Cancer''; EU FP7 MODHEP EU grant no. 259743; Labex DEEP (ANR-11-LBX-0044) part of the IDEX PSL (ANR-10-IDEX-0001-02 PSL).

## Additional information

### Funding

| Funder | Grant reference number | Author |
| --- | --- | --- |
| EU FP7 MODHEP EU | 259743 | Edith Heard |
| Labex DEEP | ANR-11-LBX-0044 | Edith Heard |
| IDEX idx PSL | ANR-10-IDEX-001-02 PSL | Edith Heard |

The funders had no role in study design, data collection and interpretation, or the decision to submit the work for publication.

### Author contributions

KA, Conception and design, Acquisition of data, Analysis and interpretation of data, Drafting or revising the article; LS, IV, TL, NS, EB, Analysis and interpretation of data, Drafting or revising the article; MB, NR, LB-R, Acquisition of data, Drafting or revising the article; EM, RS, Drafting or revising the article, Contributed unpublished essential data or reagents; C-JC, Acquisition of data, Analysis and interpretation of data, Drafting or revising the article; EH, Conception and design, Analysis and interpretation of data, Drafting or revising the article

### Author ORCIDs

Edith Heard, http://orcid.org/0000-0001-8052-7117

### Ethics

Animal experimentation: All mice used were handled with care and according to approved institutional animal care and use committee of the Institut Curie (CEEA-IC) protocols(C 75-05-18). The work has also been conducted under the approval from the French Ministry of Higher Education and Research for the use of Genetically Modified Organisms (agreement number 5549CA-I).

## Additional files

### Supplementary files

• Supplementary file 1. Summary of RNA seq data for control and maternally depleted oocytes or two-cell stage embryos. Statistical analysis of all the single oocyte or individual two-cell stage embryo RNA-seq used in this study. Datasets are available from GEO under access number GSE75054 and GSE68139.

• Supplementary file 2. Differential gene expression at the two-cell stage and oocyte stage upon loss of maternal KDM1A. The DESeq R package was used to normalize and identify the differentially expressed genes between control and mutant embryos. Genes with 0 counts in all samples were filtered out and only the 60% of the top expressed genes were used for the analysis Differentially expressed genes were identified using a minimum Log2>1 (upregulation) or <-1 (downregulation) fold change (FC) and with an adjusted p-value lower than α = 0.05.

#### Major datasets
The following datasets were generated:

| Author(s) | Year | Dataset title | Dataset URL | Database, license, and accessibility information |
|---|---|---|---|---|
| Ancelin K, Six L, Chen C, Heard E | 2015 | LSD1 is an essential regulator of the chromatin and transcriptional landscapes during the maternal-to-zygotic | http://www.ncbi.nlm.nih.gov/geo/query/acc.cgi?acc=GSE68139 | Publicly available at the NCBI Gene Expression Omnibus (Accession no: GSE68139). |
| Ancelin K, Vassilev, Chen C, Heard E | 2015 | LSD1 is an essential regulator of the chromatin and transcriptional landscapes during the maternal-to-zygotic | http://www.ncbi.nlm.nih.gov/geo/query/acc.cgi?acc=GSE75054 | Publicly available at the NCBI Gene Expression Omnibus (Accession no: GSE75054). |

The following previously published dataset was used:

| Author(s) | Year | Dataset title | Dataset URL | Database, license, and accessibility information |
|---|---|---|---|---|
| Park S-J, Komata M, Inoue F, Yamada K, Nakai K, Ohsugi M, Shirahige K | 2013 | Inferring the choreography of parental genomes during fertilization from ultralarge-scale whole-transcriptome analysis | http://trace.ddbj.nig.ac.jp/DRASearch/submission?acc=DRA001066 | Publicly available at the DDBJ Sequence Read Archive (accession no: DRA001066). |

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
