## [Decision Letter]

Thank you for submitting your work entitled "Lsd1 is an essential regulator of the chromatin and transcriptional landscapes during the maternal-to-zygotic transition" for peer review at *eLife*. Your submission has been favorably evaluated by Janet Rossant (Senior editor), a Reviewing editor, and three reviewers, one of whom, Gavin Kelsey, has agreed to share his identity.

The reviewers all agree that the work represents a significant advance but have suggested several revisions. The Reviewing editor has drafted a set of comments from the reviews to help you prepare a revised submission.

This report demonstrates a requirement for the H3K4/H3K9 demethylase KDM1A/LSD1 as a maternally provided product in zygotic genome activation in the mouse. It provides evidence that genetically depriving oocytes of LSD1 leads to embryonic arrest by the two-cell stage. Importantly, the authors show that they can phenocopy the effect by culturing embryos in the presence of an LSD1 inhibitor, demonstrating that the effect depends upon lack of catalytic activity of LSD1 in the embryo itself, rather than a compromise in the oocyte. Furthermore, they show altered levels of H3K4 and H3K9 methylation in zygotes and/or 2-cell embryos and, by RNA-seq, that there is a failure properly to activate a substantial proportion of genes that are induced at zygotic genome activation. In addition, they show up-regulation specifically of LINE1 repeats and, importantly, demonstrate this to be at the transcriptional level by RNA-FISH of nascent transcripts. Overall, this is an important finding that reveals that *Lsd1* is a maternal effect gene, and that correct regulation of H3K4 and/or H3K9 methylation is essential for correct expression of embryonic genes.

Specific comments:

1) The mechanistic connections between oocyte LSD1 deficiency and the gene expression effects in the embryo can only be speculated upon; e.g., the possible effects of H3K4 demethylation at enhancers for orderly silencing, or failure to remove repressive H3K9 methylation. And this gap is understandable, given the limitations of ChIP-seq on low numbers of cells, compounded by the reduced number of zygotes and 2-cell embryos from *Lsd/Zp3-Cre* deficient females. Specifically regarding the observed elevated LINE1 expression, the proposed link with H3K4 demethylation seems plausible (as in Fadloun et al. 2013) and potentially easier to test. Given recent advances in ChIP methods for low numbers of cells and highly abundant features such as LINEs as the target of assessment (Fadloun et al. 2013, Brind'Amour et al. 2015), a ChIP-qPCR assay at LINE1s could be done.

2) Ancelin et al. suggest that activation of Line1 elements is contributing to the increased DNA damage and developmental arrest seen in *Lsd1 Δm/wt* embryos. Only a subset of Line1 elements are active and able to retrotranspose in the genome, and different mechanisms for H3K9me3 deposition appear to be action different Line1 families. It would be informative for Ancelin et al. to extract information about which families of Line1 element are activated in *Lsd1* Δm/wt from their RNA sequencing data or through RT-qPCR. Increased Line1 jumping in the genome would require active Line1 families (A, Tf, Gf) to be upregulated.

3) The authors propose that Line1 elements might cause the DNA damage seen in *Lsd1* Δm/wt embryos. The gH2AX image shown in Figure 5 shows a lot of DNA damage in the *Lsd1* Δm/wt embryos. There would need to be a lot of active jumping of Line1 to generate this amount of DNA damage, yet Line1 transcript levels are only elevated around 2-fold. It's possible that the elevated DNA damage in *Lsd1* Δm/wt could be caused by some of the thousands of gene transcripts that are mis-regulated in these embryos rather than Line1. The authors could make this clearer. In particular, gH2AX signals normally arise during DNA replication. Maternal depletion of polycomb proteins causes defects in zygotic gene activation and impaired DNA replication in 2 cell embryos, and a similar increase in embryos with strong gH2AX staining as is reported by Ancelin et al. (see Supplementary Figure 4 in Posfai et al., 2012, Genes Dev Vol 26 p920-932). In the 2 cell embryos that Ancelin et al. show in Figure 5, does the difference in gH2AX staining between the *Lsd1* Δm/wt and control embryos primarily reflect the mutant embryos arresting or delaying progression through the mid 2 cell stage/DNA replication rather than Line1-induced DNA damage?

4) It is somewhat disappointing that one major finding namely significant effect on the second meiosis was essentially ignored (beside brief description). One would expect that the authors invest some time in trying to provide mechanistic explanation for this phenomenon and not just say that it "merits further exploration". It would be of interest to see what is happening during meiosis I and this should be included in the revised manuscript. In addition, the only significant molecular information is the transcriptome comparison between the maternally deleted and wt 2-cell stage embryos i.e. period of zygotic genome activation (ZGA). However, the ZGA is entirely regulated by the maternally inherited molecules; proteins and RNAs. Proteome comparison would be technically quite difficult but transcriptome comparison between maternally deleted and wt oocyte is possible, necessary and should be included and interpreted in the revised manuscript. Furthermore, cytoplasmic polyadenylation is one of the most important mechanisms controlling the utilization of many maternal mRNAs and it would be very important to determine how it functions during maturation of mutant oocytes and after fertilization in zygotes.

5) The authors describe a significant proportion of Δm/wt embryo arrested at zygote stage. They used superovulation and that could potentiate this effect and possibly other observed differences. It would have been much better if superovulation was not used in all experiments, however it would not be fair to ask the authors to repeat everything. Nevertheless, they could and should determine if the effect on zygote to 2-cell stage transition in Δm/wt is ameliorated when using normal matings without superovulation.

6) The authors report elevated levels of H3K9me3 in both paternal and maternal pronuclei in zygotes, elevated H3K9me1/2/3 in 2-cell embryos and elevated H3K4me1/2/3 in two cell embryos, based on immunofluorescence staining. Quantifying IF intensity is an inexact science. Although I think the conclusions are probably valid, the quantification is based on comparing IF intensity between embryos and there does not seem to be any obvious internal control. Granted, the embryos are processed in parallel and equivalent image settings are used, etc., but ideally fluorescence intensity of the modification of interest would be compared to an internal reference. For example, the authors report that levels of H3K27me3 and H4K20me3 do not differ between controls and mutant (as expected), so one or other of these modifications could be used as an internal control in dual staining for greater confidence of the changes seen (or total H3).

[Editors' note: further revisions were requested prior to acceptance, as described below.]

Thank you for resubmitting your work entitled "LSD1 is an essential regulator of the chromatin and transcriptional landscapes during the maternal-to-zygotic transition" for further consideration at *eLife*. Your revised article has been favorably evaluated by Janet Rossant (Senior editor), a Reviewing editor, and three reviewers. The manuscript has been improved but there are some remaining issues that need to be addressed before acceptance, as outlined below:

The reviewers recognise that there are additional mechanistic insights into the phenotypes that would be interesting to pursue and the editors agree that these are outside the scope of the current manuscript.

*Reviewer #1:*

The authors addressed (or attempted to address) most of the comments. They also provided some additional experiments and discussed why some of the requested experiments could not be performed. The arguments (shortage of time, being out of the scope of the investigation, limit imposed by the available number of embryos) seem overall reasonable and justified. The manuscript is of sufficient quality and interest to justify publication in *eLife*.

*Reviewer #2:*

The authors have responded appropriately and constructively to my comments and those of the other reviewers. They have added some important new data that directly address issues raised, and these new data include assessment of timing of DNA replication, and single-cell RNA-seq on *Lsd*-null oocytes, which reveals distinct transcriptional defects in oocytes and in two-cell embryos. As before, I believe that the LSD1 inhibitor experiment is an important component of the study to show that impairments in embryo progression depend upon absence of LSD1 activity in embryos, in addition to any consequence of developmental defects in oocytes themselves. There is clearly more to extract on the mechanistic side of the phenotypes described, but this will require investigations beyond the scope of this manuscript.

*Reviewer #3:*

This revised manuscript shows quite convincingly that a conditional deletion of LSD1, a histone lysine demethylase, in growing oocytes results in post-fertilization defects in pre-implantation development with the resulting embryos arresting primarily at the 2 cell stage. The authors show by immunofluorescence that histone H3K9 methylation is altered in the mutant zygotes, and that histone H3K4 and H3K9 methylation levels are altered in 2 cell embryos. The authors perform RNA seq transcriptomic analysis on mutant oocytes and 2 cell embryos that transcription profiles are also changed with the 2 cell embryos seemingly unable to activate zygotic gene activation. The authors also show that L1 retrotransposon transcripts and protein are elevated in mutant 2 cell embryos, and that these embryos also have increased amounts of the DNA damage marker gH2AX.

In response to the reviewers’ comments the authors have tried show which subtype of L1 retrotransposon is upregulated, and tested if the increased abundance of gH2AX represents a delay in cell cycle or S phase progression rather than L1 integrating into the genome as suggested in the original submission.

The authors included in their response to reviewers’ comments data showing that they could not detect upregulation of RNA for any of the three active subclasses of L1 in the LSD1 mutant embryos. The authors conclude in their response that "Nevertheless the higher levels of L1 ORF protein that we observed in mutant embryos suggest that certain L1 elements must be up-regulated in the absence of *Lsd1*, but that the LINEs at the origin of this are not systematically from one family/type". How can the authors exclude the possibility that the upregulated L1 RNA and protein represents full length but catalytically inactive L1 subfamilies such as L1_MdF? Data from ES cells shows that different L1 subfamilies are repressed epigenetic mechanisms (Castro-Diaz et al., Genes Dev. 2014 Jul 1;28(13):1397-409), so it's certainly possible that only inactive subfamilies of L1 are being upregulated in LSD1 mutant embryos. If the upregulated L1's do not originate from an active subfamily, and rather represent full length but catalytically inactive copies of L1 such as those found in the L1_MdF subfamilies, how can the increase in gH2AX be caused by L1? If the upregulated L1 copies are not catalytically active, how are they causing or contributing to the LSD1 mutant phenotype?

The authors include data showing that cell cycle or S phase is delayed in the LSD1 mutant embryos, which could cause the large increase in gH2AX seen in the mutant embryos. The authors also argue that the large amount of gH2AX in the Edu-negative mutant embryos could represent increased L1 activity. How can the authors exclude that the gH2AX in the EdU-negative embryos doesn't represent persistent DNA damage caused during S-phase? Or any other mechanism?

Although the authors have convincingly shown that maternal LSD1 is genetically required for pre-implantation development and maternal-zygotic transition, the revised manuscript does not convincingly establish mechanism for any of the developmental defects they describe in these mutants.

---

## [Author Response]

*Specific comments: 1) The mechanistic connections between oocyte LSD1 deficiency and the gene expression effects in the embryo can only be speculated upon; e.g., the possible effects of H3K4 demethylation at enhancers for orderly silencing, or failure to remove repressive H3K9 methylation. And this gap is understandable, given the limitations of ChIP-seq on low numbers of cells, compounded by the reduced number of zygotes and 2-cell embryos from Lsd/Zp3-Cre deficient females. Specifically regarding the observed elevated LINE1 expression, the proposed link with H3K4 demethylation seems plausible (as in Fadloun* et al. *2013) and potentially easier to test. Given recent advances in ChIP methods for low numbers of cells and highly abundant features such as LINEs as the target of assessment (Fadloun* et al. *2013, Brind'Amour* et al.

*2015), a ChIP-qPCR assay at LINE1s could be done.*

Although we appreciate that this is a valuable point and a fair request, unfortunately, the limited numbers of embryos that we can obtain from *Kdm1a/Lsd1* mutant female mice preclude the use of ChIP-qPCR to address defects in H3 methylationin *Lsd1*-depleted embryos. As discussed in the manuscript (Figure 1 and new Figure 1—figure supplement 2), *Kdm1a/Lsd1* deletion leads to severely reduced embryo numbers, even when using superovulation. Consequently, only very limited number of two-cell embryos could be isolated for a given experiment from each female. On average we only obtain 3 to 4 two-cell embryo per female carrying *Lsd1* conditional KO in her germline: i.e. for 206 oocytes or embryos collected from 12 females, we only obtain 20% (two-cell stage mutant embryos = 3-4 two-cell embryos per female. Thus, in order to collect even 1000 cells (the lowest numbers for which ChIP has been performed adequately – for example in germ cells – Brind'Amour, Nature Communication 2015), we would need 500 embryos which means at least 100 mutant females. Unfortunately, this is just not feasible in our animal facility and would also not be acceptable according to current animal welfare legislation. This was the primary reason why most of our analyses were carried out using immunofluorescence staining and single two-cell stage embryo RNA seq.

*2) Ancelin et al. suggest that activation of Line1 elements is contributing to the increased DNA damage and developmental arrest seen in Lsd1 Δm/wt embryos. Only a subset of Line1 elements are active and able to retrotranspose in the genome, and different mechanisms for H3K9me3 deposition appear to be action different Line1 families. It would be informative for Ancelin* et al.

*to extract information about which families of Line1 element are activated in Lsd1 Δm/wt from their RNA sequencing data or through RT-qPCR. Increased Line1 jumping in the genome would require active Line1 families (A, Tf, Gf) to be upregulated.*

Unfortunately, the single embryo RNA seq approach we have used is based on polydT priming (as originally described in Tang et al. 2010) and therefore only enables the 3’ ends of L1 transcripts to be interrogated. As the polymorphisms (unique monomer repeats), enabling different LINE-1 families to be distinguished from each other lie in their 5’UTRs, and given the length of LINEs (>6kb), this means that based on our RNA seq data we cannot determine which particular LINE-1 families were expressed in the mutants, nor whether the LINE-1 reads we detected corresponded to full length elements.

We nevertheless tried to address this important but challenging question raised by the reviewers, by adapting our single embryo RTqPCR approach using sets of LINE-1 family-specific primers for the RT step. The results are summarized in Figure 6. The aim was to produce cDNAs with internal priming rather than from the 3’UTR, in order to obtain the family-specific information from the 5’ end. We therefore designed RT primers within the LINE-1 ORF1 region, common to all families (see Figure 6). We also designed RT primers specific for two house-keeping genes that we used for normalization (*Ppia; Hprt*), and for two genes (*Btg4* and *H2AZ*) that show differential expression in our RNA seq, that we used as controls. We were thus able to evaluate expression of the 5’ ends of potentially full-length LINE-1 elements of the three families in individual control and mutant embryos. We found that only the Gf family (but not the A and Tf families), showed enhanced expression in some mutant embryos, compared to control (see Figure 6). However, the differences in Gf expression between mutant and control embryos overall were not statistically significant. We also attempted nascent RNA FISH using specific probes for each of these three LINE-1 families, but the quality of the FISH signals obtained was not adequate in embryos (when compared to ESCs), thus preventing us from making a comparison.

Author response image 1.(**A**) Ct value of qPCR with cDNA obtained using specific primers against hprt and ppia for reverse transcription.Each row corresponds to one embryo (two-cell stage). This graph shows the reproducibility and good amplification between samples. (**B**) Graphical representation of the normalized expression level (mean ± sem) as determined by qPCR for H2AZ and Btg4, two genes found respectively down and up regulated in our RNA seq. Data are expressed as normalized expression to the two house-keeping genes shown in A. (**C**) schematic representation of LINE-1 elements with the region targeted by RT specific primers and the 5' region used to design qPCR primers for each family. (**D**;**E**) qPCR analysis for LINE-1 Tf, A and Gf expression levels. The graph shows the normalized expression level for the mean ± sem (D; E left) or the normalized expression from individual embryos represented as a single bar (E right). f/wt Control are in white and Δm/wt mutant are in black.**DOI:**
http://dx.doi.org/10.7554/eLife.08851.018

In summary, we cannot conclude definitively on which LINE-1 elements are up-regulated upon loss of maternal *Lsd1* in two-cell stage embryos. In fact, it could be the case that different LINE-1 elements become spuriously over-expressed in different embryos, hence no single family shows a dramatic increase in transcription based on RT-PCR. Nevertheless, we hope the reviewers agree that the significant increase in LINE-1 ORF protein levels (both nuclear and cytoplasmic) that we detected in most *Lsd1* mutant embryos, as shown in Figure 5, is convincing evidence that full-length LINE-1 elements are indeed more frequently up-regulated in the absence of *Lsd1*.

*3) The authors propose that Line1 elements might cause the DNA damage seen in Lsd1 Δm/wt embryos. The gH2AX image shown in Figure 5 shows a lot of DNA damage in the Lsd1 Δm/wt embryos. There would need to be a lot of active jumping of Line1 to generate this amount of DNA damage, yet Line1 transcript levels are only elevated around 2-fold. It's possible that the elevated DNA damage in Lsd1 Δm/wt could be caused by some of the thousands of gene transcripts that are mis-regulated in these embryos rather than Line1. The authors could make this clearer. In particular, gH2AX signals normally arise during DNA replication. Maternal depletion of polycomb proteins causes defects in zygotic gene activation and impaired DNA replication in 2 cell embryos, and a similar increase in embryos with strong gH2AX staining as is reported by Ancelin et al.(see Supplementary Figure 4 in Posfai et al., 2012, Genes Dev Vol 26 p920-932). In the 2 cell embryos that Ancelin et al. show in Figure*

*5G, does the difference in gH2AX staining between the Lsd1 Δm/wt and control embryos primarily reflect the mutant embryos arresting or delaying progression through the mid 2 cell stage/DNA replication rather than Line1-induced DNA damage?*

We appreciate this important suggestion. As shown in our Gene Ontology pathways (GO) in Figure 4, none of the top enriched terms relate to DNA damage/ repair pathways. However, it is also plausible that enriched γH2AX staining in mutant two-cell stage embryos might be related to the many transcripts misregulated as part of the changes of the transcriptome in *Lsd1* mutants. We have now included this interpretation in the revised manuscript (at the end of the subsection “Impact of LSD1 absence on repeat elements, genome integrity and DNA replication”).

Additionally, we analysed DNA replication in mutant and control two-cell embryos (40-41h post hCG injection) using Edu pulse incorporation followed by immunostaining (Figure 5—figure supplement 2). In WT embryos, S-phase is normally complete by this time at the two-cell stage. The replication assay was coupled with parallel detection of DNA damage by γH2AX immunostaining, to address the reviewers’ suggestion that replication delay / stalling in *Lsd1* mutants might be an alternative explanation for the DNA damage observed. These new results reveal that DNA replication is indeed delayed, as replication foci are detected in a high proportion of mutant two cell stage embryos (62% blastomeres show EdU foci) compared to WT (no foci detectable). Furthermore, high levels of γH2AX foci are also observed in the mutant embryos showing EdU foci. However, we noted that in the 38% of mutant blastomeres that are not in S phase (ie Edu negative), γH2AX foci are still strongly present (compared to WT), although there may be slightly fewer foci than for the embryos in S phase. Importantly, control embryos processed in parallel are systematically negative or only faintly positive for γH2AX foci. It should be noted that γH2AX staining in this double EdU/γH2AX experiment is fainter *that*shown for γH2AX alone in Figure 5, because of differences in the IF procedure when performing the EdU assay simultaneously. Our interpretation of this new set of data is that the increase in γH2AX signalling in *Lsd1* maternally depleted embryos could indeed be linked to replication delays, as raised by the reviewers. The fact that mutant embryos not undergoing DNA replication nevertheless showed strong γH2AX foci, supports a possible correlation between DNA damage and LINE-1 overexpression. We have now modified our conclusions to evoke the different possible sources of increased DNA damage in *Lsd1* mutants, as suggested by the reviewers, in the revised manuscript (in the last paragraph of the aforementioned subsection) together with the description of new Figure 5—figure supplement 2.

*4) It is somewhat disappointing that one major finding namely significant effect on the second meiosis was essentially ignored (beside brief description). One would expect that the authors invest some time in trying to provide mechanistic explanation for this phenomenon and not just say that it "merits further exploration". It would be of interest to see what is happening during meiosis I and this should be included in the revised manuscript. In addition, the only significant molecular information is the transcriptome comparison between the maternally deleted and wt 2-cell stage embryos i.e. period of zygotic genome activation (ZGA). However, the ZGA is entirely regulated by the maternally inherited molecules; proteins and RNAs. Proteome comparison would be technically quite difficult but transcriptome comparison between maternally deleted and wt oocyte is possible, necessary and should be included and interpreted in the revised manuscript. Furthermore, cytoplasmic polyadenylation is one of the most important mechanisms controlling the utilization of many maternal mRNAs and it would be very important to determine how it functions during maturation of mutant oocytes and after fertilization in zygotes.*

We agree that the impact of loss of function of *Kdm1a /Lsd1* during oogenesis and in particular on the completion of meiosisis a very interesting question. However, our present study has focused on the early events after fertilization and already represents several years of work. We would not be able to generate new data to address the role of *Lsd1* during meiosis within a reasonable time frame for the current manuscript. In fact, this question would be part of a follow up project that we have set up in collaboration with another group who have the relevant expertise.

In the context of our manuscript, we do agree with the reviewers that the role of the maternal pools of RNA and proteins is indeed critical at the beginning of development, at the time of ZGA, and that loss of Lsd1 from the oocyte stage could impact on this maternal pool, and thus impair early development. As requested, we have now produced and analyzed transcriptomes from *Lsd1* maternally depleted and WT oocytes, as presented in Figure 4—figure supplement 2. We compared these transcriptomes to determine whether the loss of *Lsd1* function has any impact on the production of the maternal pool of transcripts in the oocyte and whether this could be at the origin of the aberrant changes detected in two-cell stage embryos.

As shown in panel A of Figure 4—figure supplement 2, fewer genes are affected (up or down regulated) in the transcriptomes of mutant oocytes (Venn diagram on the left), when compared to those of mutant two-cell embryos (Venn diagram on the right; see also Figure 4). By performing principal component analysis (PCA; Figure 4—figure supplement 2 panel B), the first dimension reveals that there is a greater difference between mutant and control two-cell stage embryos, than between mutant and control oocytes, or between maternally depleted two-cells stage embryos. However, the second dimension of the PCA shows that the mutant two-cell stage embryos segregate from oocytes, including those without a maternal *Lsd1* supply. Finally, the number of identically misregulated genes (up or down) between oocytes and two-cell stage embryos (panel C) is far lower than the number of misregulated genes that are specific to each stage. Furthermore, they do not match extensively with annotated maternal genes (based on www.dbtmee.hgc.jp; see also Figure 5). Thus, according to this new data, loss of *Kdm1a/Lsd1* leads to more genes being misregulated *after* fertilization, than in oocytes. In other words, *Lsd1* depletion seems to affect the transcriptome of the developing embryo more than it does that of the maturating oocyte. Our new oocyte transcriptome data thus strengthen our original conclusion that *Lsd1* is indeed required to shape the two waves of ZGA after fertilization, from zygote to two-cell stage. However, we cannot exclude that an *Lsd1*-depleted maternal pool would be deficient for some maternal RNAs or proteins (e.g. transcription factors), that could impact transcription at the two-cell stage and have included a comment to this effect in our revised manuscript (subsection “Absence of LSD1 abrogates the normal changes in transcriptome by the two-cell stage”, fourth paragraph).

Finally, we appreciate the reviewers’ comment that regulation of cytoplasmic polyadenylation might also be affected in the *Kdm1a/Lsd1* mutant, as this plays an important role in the use of the maternal RNA supply. However, a study of poly-A length in *Lsd1* mutants was hindered due to the use of polydT primers for production of our RNA seq data. We do however now mention the possibility that this might also play a role (subsection “LSD1 is involved in the transcriptional switch at the two-cell stage”).

*5) The authors describe a significant proportion of Δm/wt embryo arrested at zygote stage. They used superovulation and that could potentiate this effect and possibly other observed differences. It would have been much better if superovulation was not used in all experiments, however it would not be fair to ask the authors to repeat everything. Nevertheless, they could and should determine if the effect on zygote to 2-cell stage transition in Δm/wt is ameliorated when using normal matings without superovulation.*

Superovulation was used in our study because of the very limited number of mutant embryos we could obtain (as shown in Figure 1). We nevertheless agree with the reviewers that we should rule out any possible effect that superovulation might have on the developmental delay seen at the two-cell stage for the *Kdm1a/Lsd1* mutant embryos. We addressed this issue by assessing the development of mutant and control embryos obtained by natural breeding, without superovulation. The data are provided as a new supplemental figure (Figure 1—figure supplement 2) in the revised manuscript. The number of embryos recovered for each female was low (as expected without superovulation), but the proportion of one-cell and two-cell embryos was not markedly different from what we had obtained using superovulation. In conclusion, superovulation did not change our conclusions concerning *Lsd1* mutant embryo phenotypes.

*6) The authors report elevated levels of H3K9me3 in both paternal and maternal pronuclei in zygotes, elevated H3K9me1/2/3 in 2-cell embryos and elevated H3K4me1/2/3 in two cell embryos, based on immunofluorescence staining. Quantifying IF intensity is an inexact science. Although I think the conclusions are probably valid, the quantification is based on comparing IF intensity between embryos and there does not seem to be any obvious internal control. Granted, the embryos are processed in parallel and equivalent image settings are used, etc., but ideally fluorescence intensity of the modification of interest would be compared to an internal reference. For example, the authors report that levels of H3K27me3 and H4K20me3 do not differ between controls and mutant (as expected), so one or other of these modifications could be used as an internal control in dual staining for greater confidence of the changes seen (or total H3).*

We agree with the reviewers’ comment that IF quantifications can be subject to signal intensity variation, between samples, or between experiments. However, we have done our utmost to control for such variability in this work. As mentioned in our Materials and methods section (subsections “Immunofluorescence staining” and “Confocal acquisition and image analysis”), for each antibody used, immunofluorescence was always performed and processed on embryos from control and mutant genotypes in parallel, and under identical conditions. Images were always acquired using the same confocal microscope settings, even between experiments.

In terms of quantification, and the use of an internal control, we believe that performing dual IF with a “control” antibody is not necessarily ideal, as this is also subject to variation between fluorochromes from one colour to another. Instead, we believe that our approach allows us to deduce whether there are significant differences in overall staining intensities but we realise we may not have adequately described this in our manuscript. In summary, we select the nuclear area of the stack image, and then extract the integrated Intensity (intensity divided by the number of voxels represented within the nuclear area) using the 3D object counter plugin in Image J (Bolte and Cordelière, 2006). This is now added in our Materials and methods section (subsection “Confocal acquisition and image analysis”).

In order to show the raw data of fluorescence measurements, we provide a figure (Figure 7) in which are presented the distributions of the integrated intensities measured per nucleus for each of the embryos (control or mutant) stained with anti-H3K4me3, anti-H3K9me3 or anti-H3K27me3. These raw data were used to produce the ratio of changes between control and mutant embryos plotted in Figure 3 or Figure 3—figure supplement 1, under each image panel.

The three graphs in Figure 7 show that, although there is some variability for each antibody used between nuclei, and between experiments (we performed at least two replicates), for both the control and the mutant categories, the distribution of the intensities are significantly different between control and mutant embryos for H3K4me3 and for H3K9me3, but not for H3K27me3, (based on use of a t-test, as presented in the Figure 3).

Author response image 2.**DOI:**
http://dx.doi.org/10.7554/eLife.08851.019

[Editors' note: further revisions were requested prior to acceptance, as described below.]

*Reviewer #3: This revised manuscript shows quite convincingly that a conditional deletion of LSD1, a histone lysine demethylase, in growing oocytes results in post-fertilization defects in pre-implantation development with the resulting embryos arresting primarily at the 2 cell stage. The authors show by immunofluorescence that histone H3K9 methylation is altered in the mutant zygotes, and that histone H3K4 and H3K9 methylation levels are altered in 2 cell embryos. The authors perform RNA seq transcriptomic analysis on mutant oocytes and 2 cell embryos that transcription profiles are also changed with the 2 cell embryos seemingly unable to activate zygotic gene activation. The authors also show that L1 retrotransposon transcripts and protein are elevated in mutant 2 cell embryos, and that these embryos also have increased amounts of the DNA damage marker gH2AX. In response to the reviewers comments the authors have tried show which subtype of L1 retrotransposon is upregulated, and tested if the increased abundance of gH2AX represents a delay in cell cycle or S phase progression rather than L1 integrating into the genome as suggested in the original submission. The authors included in their response to reviewers’ comments data showing that they could not detect upregulation of RNA for any of the three active subclasses of L1 in the LSD1 mutant embryos. The authors conclude in their response that "Nevertheless the higher levels of L1 ORF protein that we observed in mutant embryos suggest that certain L1 elements must be up-regulated in the absence of Lsd1, but that the LINEs at the origin of this are not systematically from one family/type". How can the authors exclude the possibility that the upregulated L1 RNA and protein represents full length but catalytically inactive L1 subfamilies such as L1_MdF? Data from ES cells shows that different L1 subfamilies are repressed epigenetic mechanisms (Castro-Diaz et al., Genes Dev. 2014 Jul 1;28(13):1397-409), so it's certainly possible that only inactive subfamilies of L1 are being upregulated in LSD1 mutant embryos.*

We agree with the reviewer that this is a possibility that we cannot formally rule out – although it seems rather unlikely as we do not see any specific L1 subfamily expressed (L1_MdF or other). We have now included a new sentence in the Results section: “in particular we could not determine which specific LINE-1 families were expressed in the mutants, nor whether the LINE-1 reads we detected corresponded to full-length, and/or intact elements.”

*If the upregulated L1's do not originate from an active subfamily, and rather represent full length but catalytically inactive copies of L1 such as those found in the L1_MdF subfamilies, how can the increase in gH2AX be caused by L1?*

In our Discussion we have tried to be very cautious concerning any causality between LINE-1 and gH2AX signalling: “although, no significant enrichment was directly found for DNA damage pathways when running our GO analysis (Figure 4), many genes related to DNA damage repair were upregulated ([Supplementary-material SD1-data]). Taken all together, these results suggest that the elevated DNA damage signalling observed could be independent from replication defaults in KDM1A maternally depleted embryos, but might be related either to changes in transcript levels for DNA damage genes or else to the observed increase in LINE-1 activityin *Kdm1a* mutant embryos at this stage”.

*If the upregulated L1 copies are not catalytically active, how are they causing or contributing to the LSD1 mutant phenotype?*

Our work suggests that KDM1A may participate in repressing LINE-1 expression and that in its absence we find increased protein levels, likely due to partial up-regulation of different types of L1 elements. Although some of these may produce catalytically inactive protein, we have no reason to exclude that a subset of them are active and might indeed result in genome instability. However this has not been formally demonstrated and we fully acknowledge that the DNA damage observed in the mutant embryos may be due to other defects, including DNA replication defects. We have therefore included the following statement which we hope satisfies the reviewer “In this context, we speculate that misregulation of LINE-1 elements in the absence of KDM1A *might participate* in the early developmental arrest that is observed, via an increased potential of genome instability and activation of some specific DNA damage checkpoints”.

*The authors include data showing that cell cycle or S phase is delayed in the LSD1 mutant embryos, which could cause the large increase in gH2AX seen in the mutant embryos. The authors also argue that the large amount of gH2AX in the Edu-negative mutant embryos could represent increased L1 activity. How can the authors exclude that the gH2AX in the EdU-negative embryos doesn't represent persistent DNA damage caused during S-phase? Or any other mechanism?*

We do not exclude that the DNA damage observed is caused in part by replication, we simply evoke the possibility that it could also be linked to LINE-1 activation, as mentioned in the statement in answer to the previous point.

Although the authors have convincingly shown that maternal LSD1 is genetically required for pre-implantation development and maternal-zygotic transition, the revised manuscript does not convincingly establish mechanism for any of the developmental defects they describe in these mutants.

Although we have not established detailed mechanism, we do hope that the reviewer agrees that we have definitively shown the importance of the maternal KDM1A protein for early life, including its role in ensuring appropriate H3K4 and H3K9methylation patterns and in early transitions in gene expression. We believe that this work opens up the way for the further exploration of the detailed mechanisms by which this important epigenetic regulator works at multiple levels.